# Structural differences between the closely related RNA helicases, UAP56 and URH49, fashion distinct functional apo-complexes

Ken-ichi Fujita[1,2,3] ✉, Misa Ito[1], Midori Irie[1], Kotaro Harada[1], Naoko Fujiwara[1], Yuya Ikeda[1], Hanae Yoshioka[1], Tomohiro Yamazaki [1], Masaki Kojima [4], Bunzo Mikami[5,6], Akila Mayeda [2] & Seiji Masuda [1,7,8,9] ✉

mRNA export is an essential pathway for the regulation of gene expression. In humans, closely related RNA helicases, UAP56 and URH49, shape selective mRNA export pathways through the formation of distinct complexes, known as apo-TREX and apo-AREX complexes, and their subsequent remodeling into similar ATP-bound complexes. Therefore, defining the unidentified components of the apo-AREX complex and elucidating the molecular mechanisms underlying the formation of distinct apo-complexes is key to understanding their functional divergence. In this study, we identify additional apo-AREX components physically and functionally associated with URH49. Furthermore, by comparing the structures of UAP56 and URH49 and performing an integrated analysis of their chimeric mutants, we exhibit unique structural features that would contribute to the formation of their respective complexes. This study provides insights into the specific structural and functional diversification of these two helicases that diverged from the common ancestral gene Sub2.

During the expression of protein-coding genes, pre-mRNAs are transcribed in the nucleus and undergo several RNA processing steps, including capping, splicing, and polyadenylation. Subsequently, the mature mRNA is exported to the cytoplasm for translation. These processes are coupled with one another through appropriate assembly and remodeling of mRNA-protein (mRNP) complexes to achieve accurate gene expression[1].

A key player integrating transcription and mRNA export is the evolutionarily conserved ATP-dependent multi-subunit TRanscription-EXport (TREX) complex. The human ATP-bound TREX complex consists of the THO subcomplex, comprising THOC1, THOC2, THOC3,

THOC5, THOC6, and THOC7, and several affiliated proteins: ALYREF, CIP29, CHTOP, PDIP3, ZC3H11A, and DEAD-box RNA helicase UAP56 (also called DDX39B, and references therein[4–6]). The TREX components are recruited onto transcribing RNA polymerase II (Pol II) and loaded onto spliced-mRNA in a splicing-dependent manner, which is crucial for subsequent export[5,7].

Perhaps the most crucial factor in the assembly of the TREX complex is UAP56 (Sub2 in yeast). During splicing, UAP56 is loaded onto pre-mRNA through the interaction with U2AF65[8], and in turn, it regulates spliceosome assembly[9,10]. UAP56 interacts with the THO subcomplex in an ATP-independent manner[11]. When ATP binds UAP56,

[1]Division of Integrated Life Sciences, Graduate School of Biostudies, Kyoto University, Kyoto, Kyoto 606-8502, Japan. [2]Division of Gene Expression Mechanism, Center for Medical Science, Fujita Health University, Toyoake, Aichi 470-1192, Japan. [3]Division of Cancer Stem Cell, National Cancer Center Research Institute, 5-1-1 Tsukiji, Chuo-ku, Tokyo 104-0045, Japan. [4]School of Life Sciences, Tokyo University of Pharmacy and Life Sciences, Hachioji, Tokyo 192-0392, Japan. [5]Research Institute for Sustainable Humanosphere, Kyoto University, Kyoto 611-0011, Japan. [6]Institute of Advanced Energy, Kyoto University, Kyoto 611-0011, Japan. [7]Department of Food Science and Nutrition, Faculty of Agriculture Kindai University, Nara, Nara 631-8505, Japan. [8]Agricultural Technology and Innovation Research Institute, Kindai University, Nara, Nara 631-8505, Japan. [9]Antiaging Center, Kindai University, Higashiosaka, Osaka 577-8502, Japan. ✉e-mail: kefujit@ncc.go.jp; smasuda@nara.kindai.ac.jp

it recruits CIP29, ALYREF, CHTOP, PDIP3, and ZC3H11A into the TREX complex[2,3,6]. Thus, the TREX complex is remodeled from the ATP-unbound TREX complex to the ATP-bound TREX complex via ATP binding to UAP56. We term the ATP-unbound form as the apo-TREX complex and the ATP-bound one as the ATP-TREX complex to distinguish both complexes[11] (Fig. 1A). The formation of the ATP-TREX complex drives the export of bound mRNA because ALYREF, CHTOP, and THOC5 act as adapters of the NXF1-NXT1 heterodimer which functions in the final step of the mRNA export[2,12,13].

In mammals, UAP56 has a paralogue that is 90% identical to URH49[14] (also called DDX39A). Furthermore, we have previously shown that UAP56 and URH49 form distinct apo-complexes. UAP56 forms the apo-TREX complex, and URH49 forms the apo-Alternative-mRNA-EXport (AREX) complex. Unlike the apo-TREX complex, the apo-AREX complex contains CIP29 and it does not contain the THO subcomplex[15]. Like the apo-TREX complex, the apo-AREX complex is remodeled to ATP-complex when ATP is loaded onto URH49 and accesses NXF1-NXT1 heterodimer for mRNA export[11]. Irrespective of whether the precursor is an apo-TREX or an apo-AREX complex, ATP complexes resemble each other and are called the ATP-TREX complex.

Each helicase selectively exports a distinct subset of mRNAs, including key mitotic regulators[15]. URH49 is also required for the gene expression involved in cytokinesis[11]. In addition, abnormalities in their mRNA export pathways, including disrupted expression, have been associated with serious diseases, such as cancer and neurodegenerative disorders[7,16]. Thus, the evolutionarily diversified mRNA export pathways formed by UAP56 and URH49, along with their respective complex components, contribute to the fine-tuned gene expression and are required for a variety of physiological events[17–19]. Therefore, elucidation of the functional machinery of UAP56 and URH49 and their differences is important, not only for a better understanding of gene regulation in higher organisms but also for an understanding of a variety of diseases caused by the disruption of these two helicases.

DEAD-box family helicases generally bind RNA in a sequence-independent manner, and target recognition is primarily provided via partner proteins[20]. Thus, identifying the compositions of the apo-TREX and the apo-AREX complexes, and elucidating the molecular basis of the involvement of UAP56 and URH49 in complex formation, may be the key to understanding their function. However, the factor(s) of the apo-AREX complex are unknown except for CIP29, which is also in the ATP-TREX complex (Fig. 1A). Another important aspect is that, despite their extensive homology, the mechanisms by which UAP56 and URH49 form distinct complexes remains unknown.

In this study, we first used tandem immunoprecipitation and mass spectrometry to investigate the factors of the apo-AREX complex. Then, we determined the reason why UAP56 and URH49 form different apo-complexes.

## Results

### Identification of the apo-AREX components

To analyze the composition of the apo-AREX complex, we performed immunoprecipitation using RNase-treated nuclear extract prepared from Flp-In T-REx 293 cells expressing either FLAG-UAP56 or FLAG-URH49 in the ATP-depleted condition. FLAG-UAP56 or FLAG-URH49 coprecipitated different components: FLAG-UAP56 was associated with the apo-TREX components (THOC1, THOC2, and THOC5), and FLAG-URH49 precipitated the apo-AREX component CIP29 (Fig. 1B). These interactions are consistent with previously reported different interactions of endogenous UAP56 and URH49[11,15]. We then added ATP and found that the ATP-TREX components (THOC1, THOC2, THOC5, ALYREF, and CIP29) interacted with both FLAG-UAP56 and FLAG-URH49. We refer to the ATP-TREX complex containing UAP56 as the ATP-TREX (UAP56) complex and the ATP-TREX complex containing URH49 as the ATP-TREX (URH49) complex. This remodeling was also observed in the presence of ADP or AMP-PNP (Fig. 1B, Supplementary

Fig. 1A). There was a slight difference in the composition of the ATP-TREX (URH49) complex in the presence of ADP and AMP-PNP. UAP56 binds AMP-PNP with affinities at least 10 times lower than that of ATP and binds ADP with similar affinities as ATP[21]. Therefore, with the addition of AMP-PNP, the URH49 immunoprecipitates may contain both the ATP-TREX (URH49) complex and, partially, the apo-AREX complex formed by apo-URH49 molecules that do not bind to AMP-PNP. Under conditions of excess AMP-PNP, immunoprecipitation of URH49 revealed coprecipitates closely resembling those of ATP-bound URH49 (see Supplementary Fig. 1B). These results indicated that the ATP binding, but not the ATP hydrolysis is sufficient to exert the complex remodeling. In the FLAG-URH49 precipitate, we detected many candidates for the apo-AREX components. To identify these factors as authentic apo-AREX components, we performed tandem purification of the apo-AREX complex with the nuclear extract expressing both known apo-AREX components, FLAG-URH49 and HA-CIP29, and identified isolated factors by LC-MS/MS (Fig. 1C, Supplementary Fig. 1C–E).

Among the coimmunoprecipitated factors of FLAG-URH49 and FLAG-UAP56, we observed enrichment of splicing-associated factors, indicating that involvements of both helicases with the splicing process (Supplementary Fig. 2A–C, see also Supplementary Table 1). Moreover, various RNA-binding proteins are found to bind to URH49 but not UAP56. RUVBL1, RUVBL2, ILF2, ILF3, and HNRNPM were reliably detected from the tandem immunoprecipitate as well as from the FLAG-URH49 precipitate, but not in the FLAG-UAP56 precipitate (Fig. 1C, Supplementary Fig. 2B). RUVBL1 and RUVBL2 form heterodimers, as do ILF2 and ILF3[22,23]. HNRNPM interacts with ILF2, ILF3, and other factors[24]. Thus, we focused on these factors, and interactions between these factors and URH49 were confirmed by immunoblotting of FLAG-URH49 precipitate (Fig. 1D).

To confirm this, we generated cell lines stably expressing FLAG-RUVBL1, RUVBL2, ILF2, ILF3, and HNRNPM and investigated their interactions with URH49. Anti-FLAG- precipitates from nuclear extracts expressing each apo-AREX candidate efficiently captured URH49, whereas there was no obvious binding to THOC1 and ALYREF, components of the apo- and the ATP-TREX complex, was observed (Fig. 1E−G). Please see Supplementary Fig. 2D, in which complex each factor is present. The interactions between RUVBL1 and RUVBL2, as well as between ILF2 and ILF3, were sustained in the presence of ATP, as previously reported[22,23]. In contrast, these factors dissociated from URH49 upon ATP addition. These results indicate that these factors interact with URH49 as the apo-AREX complex but do not interact with the ATP-TREX complex (Supplementary Fig. 2D).

### The apo-AREX components are associated with URH49-target mRNA processing and export

Next, we evaluated the functional significance of apo-AREX candidates in mRNA processing and export. In addition to ILF3 (also known as NF110), NF90, a truncated isoform of ILF3, also interacts with ILF2[22]. A previous study reported that the use of an ILF3-specific siRNA could deplete ILF3 expression without affecting NF90 expression, while the knockdown of ILF2 downregulated the expression of both ILF3 and NF90[22]. The depletion of either RUVBL1 or RUVBL2 causes a co-depletion of the other[25]. Thus, we depleted RUVBL1, ILF3, HNRNPM, and CIP29, a known apo-AREX component by siRNA-mediated knockdown (Supplementary Fig. 3A). Depletion of either factor induced the nuclear accumulation of poly(A)$^+$ RNAs, which co-localized with nuclear speckles (Fig. 2A−C). This observation probably reflects the perturbed mRNA splicing and export by knocking down either factor, as shown by previous reports[26–28]. Similar results were observed with other cell lines and with other siRNAs of any of the factors (Supplementary Fig. 3B−D). These observations indicated that our apo-AREX candidates function in mRNA processing and export.

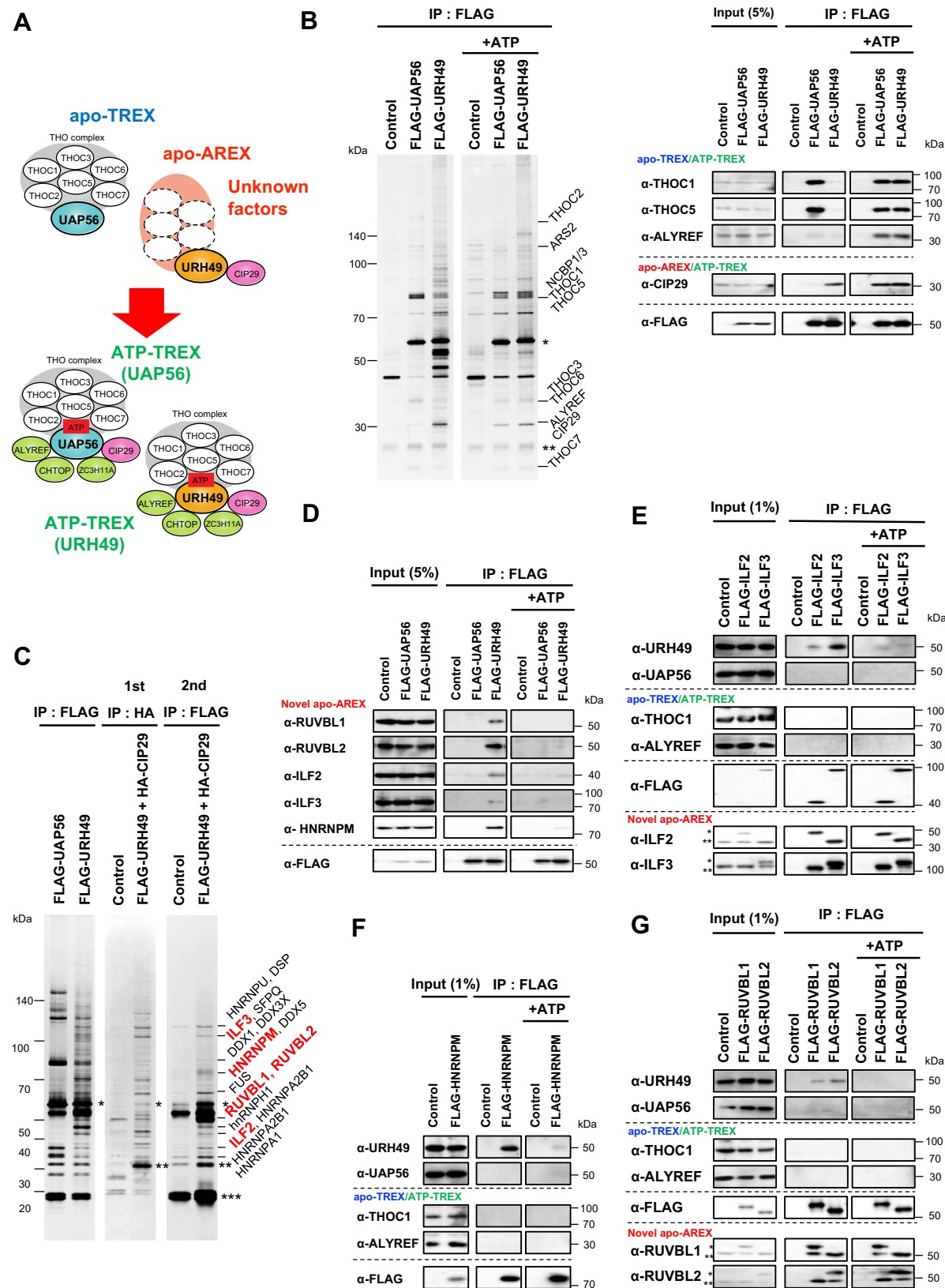

To further clarify the function of these factors in the apo-AREX complex, we depleted each apo-AREX candidate and assessed its effect on the cytoplasmic mRNA expression of UAP56 or URH49 targets. We performed reverse transcription-polymerase chain reaction (RT-PCR) after nuclear and cytoplasmic fractionation and confirmed their success (Supplementary Fig. 3E). Depletion of each factor specifically

reduced the expression of the URH49 targets but did not cause a reduction of the UAP56 targets, indeed in some cases it upregulated them (Fig. 2D). Such upregulation of UAP56 targets was also observed when URH49 and CIP29 were depleted[11], suggesting that depletion of the apo-AREX component probably enhances the UAP56 export pathway as a compensatory mechanism. These results indicated that

**Fig. 1 | Identification of apo-AREX components.** Immunoprecipitation using anti-DYKDDDDK (equivalent to FLAG) tag antibody beads to pull down nuclear extract of Flp-In T-REx 293 cells stably expressing FLAG-tagged each protein. Each precipitated sample was separated and detected by silver staining (left) or immunoblotting (right) with the indicated antibodies. **A** The model of the apo-TREX, -AREX and the ATP-TREX complex formations of UAP56 and URH49. The dotted circles refer to apo-AREX complex components that were unidentified prior to this study. **B** FLAG-UAP56 and -URH49 differ in apo-complex formation but are similar in ATP-complex formation. In the right panel, known apo-TREX, -AREX, and ATP-TREX components were detected by immunoblotting. Single asterisk represents precipitated FLAG-UAP56 or -URH49, double asterisk represents IgG light chain. **C** Identification of apo-AREX components by tandem-immunoprecipitation. Precipitated proteins by FLAG immunoprecipitation or tandem immunoprecipitation (first: HA, second: FLAG) are detected, respectively. Identified proteins are shown

on the right side. Details of proteins identified are indicated in Supplementary Fig. 2B and Supplementary Table 1. The proteins shown in the red letter were further analyzed. Single, double, and triple asterisk represented precipitated FLAG-UAP56 or -URH49, HA-CIP29, and IgG light chain, respectively. **D** FLAG-URH49 is specifically associated with each apo-AREX component in an ATP-depleted condition but not in the presence of ATP. **E** FLAG-ILF2 and -ILF3 bind with apo-URH49 in an ATP-deficient condition. Single and double asterisks represent FLAG- and endogenous-ILF2 or ILF3, respectively. **F** FLAG-HNRNPM associates with URH49 in the absence of ATP. **G** FLAG-RUVBL1 and -RUVBL2 associate with URH49 in an ATP-depleted condition. Single and double asterisks represent FLAG- and endogenous-endogenous RUVBL1 or RUVBL2, respectively. For panels (**B**–**G**), similar results were obtained in at least three independent experimental settings. Source data are provided as a Source Data file.

each factor functions as the apo-AREX complex and specifically regulates URH49-target mRNA export.

## A single amino acid difference between UAP56 and URH49 impacts apo-complex formation and function

Subsequently, we investigated how UAP56 and URH49 form distinct apo-complexes despite their high homology. DEAD-box RNA helicase contains a conserved core region with two domains (N-domain and C-domain, respectively), a linker region between them, and terminal regions[20] (Fig. 3A, Supplementary Fig. 4). We hypothesized that differences in a specific region(s) between UAP56 and URH49 are important to form their distinct apo-complexes. To identify the region(s) within UAP56 and URH49 responsible for forming different apo-complexes, we constructed plasmids expressing various mutants of UAP56 and URH49, in which different regions were swapped. We examined the formation of apo-complexes (Fig. 3A, Supplementary Fig. 5). The N- and C-terminal regions are relatively different compared to the core regions of UAP56 and URH49. However, swapping of either terminal region did not affect the apo-complex formation (Supplementary Fig. 5A). In contrast, mutants swapped of each N-domain (described as "UAP56 N-core" and "URH49 N-core") dramatically altered apo-complex formation (Supplementary Fig. 5B, C), suggesting the N-domain determines which apo-complex forms.

In humans, UAP56 and URH49 have 12 amino acid differences in the N-domain (Fig. 3A, Supplementary Fig. 4). Subsequently, we analyzed 12 point mutants in which different individual amino acids are swapped. Strikingly we found that the URH49 C223V mutant specifically switched the complex formation from the apo-AREX complex to the apo-TREX complex (Fig. 3B, Supplementary Fig. 5D). UAP56 V224C, the mutant corresponding to URH49 C223V, did not alter the apo-complex formation (Supplementary Fig. 5D). To further investigate the potential contribution of amino acid differences other than UAP56-V224 and URH49-C223 to their distinct complex formation, we generated a mutant termed "UAP56 N-core C224V." In this mutant, the N-domain of UAP56, excluding V224, was replaced with the N-domain of URH49. This mutant lost the ability to form the apo-AREX complex but retained the ability to form the apo-TREX complex (Supplementary Fig. 5E). Additionally, we selected 10 vertebrate species and aligned the amino acid sequences of their respective UAP56 and URH49 homologs (Supplementary Figs. 6 and 7). UAP56-V224 and URH49-C223 were conserved across these organisms. These results indicate that the difference between UAP56-V224 and URH49-C223 is the crucial determinant of apo-complex formation and has potential implications for the evolutionary functional divergence of UAP56 and URH49.

It has been reported that UAP56 and URH49 export not only mRNAs but also circular RNAs, which are generated via "back-splicing"[29]. In that report, UAP56 plays a role in long circular RNA export while URH49 exports short circular RNAs. The four different amino acids that are located in their N-domains between UAP56 and URH49 determine their specificity for circular RNAs. However, we did

not find a difference in complex formation between the apo-TREX and the apo-AREX in the mutants with these four amino acid substitutions (Supplementary Fig. 5F). Thus, how UAP56 and URH49 differentially regulate the export of various circRNAs is unlikely to be explained by differences in the binding partners involved in mRNA export.

We next examined whether alteration of the apo-complex formation affects mRNA export activities of the two helicases. The depletion of UAP56 or URH49 induced bulk nuclear poly(A)$^+$ RNA accumulation, respectively[15,30]. The forced expression of siRNA-resistant UAP56 rescued the nuclear poly(A)$^+$ RNA accumulation induced by endogenous UAP56 knockdown but did not rescue the nuclear poly(A)$^+$ RNA accumulation provoked by the disruption of URH49, and vice versa (Fig. 3C, D, Supplementary Fig. 8). These results reflect that UAP56 and URH49 are involved in the export of distinct subsets of mRNA substrates and that there are likely non-redundant mRNA substrates that are specific to each of them[15]. URH49 C223V and URH49 chimera mutant, URH49 N-core, which form the apo-TREX complex, could specifically rescue the nuclear poly(A)$^+$ RNA accumulation caused by the knockdown of endogenous UAP56. In addition, UAP56 chimera mutant UAP56 N-core, which forms the apo-AREX complex, could rescue the nuclear poly(A)$^+$ RNA accumulation induced by the disruption of endogenous URH49. These data clearly demonstrate that mRNA export selectivity was controlled at the apo-complex formation step. Taken together, the formation of distinct apo-complex due to the difference in a single amino acid between UAP56 and URH49 has a key role in the selective mRNA export by the two helicases.

## UAP56 and URH49 form different apo-structures but with similar ADP or ATP-binding structures

DEAD-box helicases have similar structural features[20]. In the apo-state, DEAD-box family proteins adopt a variety of open structures with the configuration of N-domain and C-domain different for each member. However, upon ATP binding, the N- and C-domains undergo rearrangement into similar closed structures driven by interactions with ATP. These structural features determine what kind of complex forms according to their apo- and ATP-binding state. In fact, the remodeling of the apo-AREX complex to the ATP-TREX complex (URH49) dramatically altered the protein composition between the two complexes (Fig. 1D–G). These results led us to hypothesize that the structures of UAP56 and URH49 in their apo- and ATP-bound states determine the formation of their apo- and ATP complexes. Supplying ADP caused the complex remodeling of UAP56 and URH49 as well as the addition of AMP-PNP, a non-hydrolyzable analog of ATP (Supplementary Fig. 1A), indicating that the ADP-bound structures of two helicases resemble the ATP-bound structures. Thus, we compared the structural features of UAP56 and URH49 in the apo- and the ADP-bound states by limited proteolysis.

The sites of the primary amino acid sequence predicted to be digested by trypsin were the same in both helicases. However, the

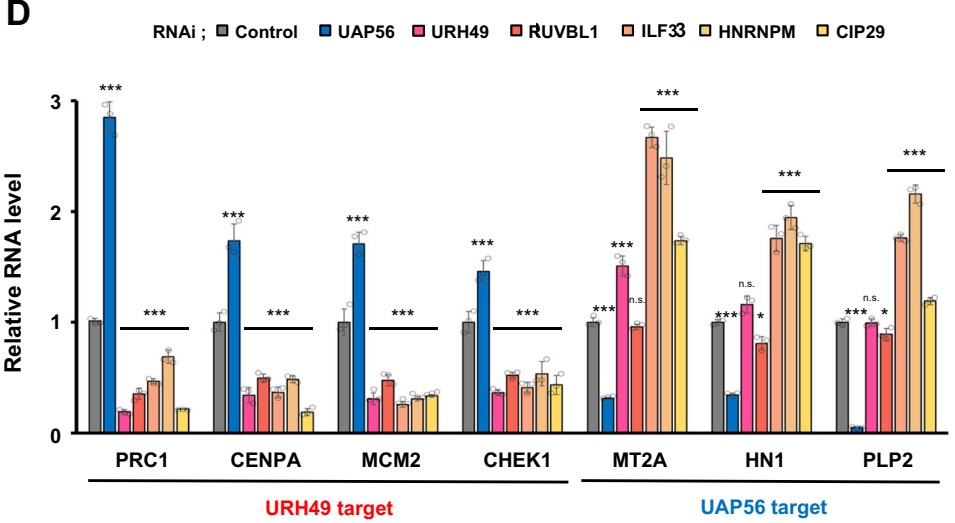

digested fragments and, thus, the structures in the apo-state differed between UAP56 and URH49 (Fig. 4A, B). As for UAP56, the digested products in the presence of ADP were similar to those in the absence of ADP. Previously, crystal structures of both apo- and ADP-bound form UAP56ΔN42, which lacked N terminal 42 residues of UAP56, were reported[31]. In that study, UAP56ΔN42 exhibits a relatively closed conformation in the apo-state compared to other DEAD-box proteins[31]. ADP binding induces a slight structural

rearrangement only around the ATP-binding pocket without the configurational change of their N- and C-domains. Our observations seem to reflect these findings. On the contrary, the digestion pattern of URH49 in the presence of ADP differed from that in the absence of ADP and changed to that of UAP56 upon the addition of ADP. Moreover, the fragments digested by UAP56 and URH49 in the presence of ATP closely resembled those digested with ADP (Supplementary Fig. 9A). This indicates that, unlike UAP56, URH49

**Fig. 2 | The apo-AREX components are specifically associated with URH49-mediated mRNA processing and export. A** Depletion of apo-AREX components caused nuclear poly(A)$^+$ RNA accumulation in U2OS cells. DAPI was used to visualize the nuclei. Scale bar, 40 μm. **B** Quantification of the nuclear poly(A)$^+$ RNA accumulation caused by the knockdown of apo-AREX components. The graph indicates the fold changes in the ratio of nuclear per cytoplasmic distribution of mRNA. These data were normalized to the score of the control knockdown condition. The signal intensities of bulk poly(A)$^+$ RNA in the nucleus and the cytoplasm were quantified using ImageJ ($n = 40$ cells of each, respectively). Boxes show the median (centerline) and upper and lower quartiles. Whiskers show the lowest and highest values. Statistical analysis was performed using the Kruskal-Wallis test followed by the Steel test. ***$p < 0.001$. **C** Localization of poly(A)$^+$ RNA in U2OS cells. Poly(A)$^+$ RNA localization (red) was observed under the knockdown of each apo-AREX component. Anti-SRRM2 antibody was used to stain the nuclear speckle (green). DAPI was used to visualize the nuclei (blue). Scale bar, 10 μm. In the right panels, signal intensities of poly(A)$^+$ RNA and SRRM2 (same colors) were plotted between the A and B lines in the left panels. **D** Depletion of apo-AREX components resulted in the decreased expression of URH49-target mRNAs in the cytoplasm. RT-qPCR was performed using the cytoplasmic RNA to compare the mRNA expression level. Values represent the relative expression of indicated mRNA normalized to PGK and the mean ± SEM of three independent experiments. Statistical analysis was performed using one-way ANOVA followed by Dunnett's test. **$p < 0.01$, ***$p < 0.001$. n.s.: not significant. For panels (**A–D**), similar results were obtained in at least three independent experimental settings. Source data are provided with details of statistical tests and exact $p$-values as a Source data file.

undergoes a significant conformational change upon binding of ADP or ATP.

We also generated the UAP56ΔN42 and URH49ΔN41, which is the URH49 mutant corresponding to UAP56ΔN42. Their digested fragments were different under the apo-condition and became similar in the presence of ADP as well as in the case of UAP56 and URH49 (Fig. 4C, D). These results indicate that UAP56ΔN42 has a similar structure to UAP56 and URH49ΔN41 to URH49. We next estimated the cleavage sites of the UAP56ΔN42 and URH49ΔN41 by detecting peptides using LC-MS/MS analysis (described A1-4 and R1-4 in Fig. 4C, D). The A1 and A2 fragments generated in the absence of ADP covered peptides from the N-domain and linker region of UAP56ΔN42. The peptide composition of the R1 fragment was similar to that of the A1 fragment, while the peptide composition of the R2 fragment contained the C-domain and the linker region and was completely different from that of the A2 fragment (Fig. 4C, E, see also Supplementary Table 2). These results indicated that the sensitive site of UAP56 digestion by trypsin was different from that of URH49, probably based on their distinct structures in the absence of ADP or ATP (Fig. 4E bottom). The A3 and 4 fragments generated in the presence of ADP had the same digestion pattern with the R3 and 4 fragments (Fig. 4D, F), indicating that URH49 underwent a significant structural change by the loading of ADP and UAP56 did not.

To further confirm that URH49 underwent the structural rearrangement upon ADP-binding, we employed mutants lacking the ATP-binding activities: UAP56ΔN42 K95N and URH49ΔN41 K94N[11]. These mutants had the same digestion pattern as one another in the presence of ADP (Supplementary Fig. 9B). From these results, we concluded that the two helicases form different apo-structures but were remodeled to similar structures on ADP binding. Importantly, the digested pattern of URH49 C223VΔN41 was similar to that of UAP56ΔN42 (Fig. 4C). This indicates that V224 of UAP56 and C223 of URH49 play important roles in forming their different apo-structures, and raises the possibility that the structural feature of UAP56 and URH49 were associated with their apo- and ADP-/ATP-complex formation.

### Structural differences between the apo-UAP56 and URH49

To analyze the difference between both apo-structures, we solved the crystal structure of URH49ΔN41 by x-ray diffraction (8IJU) and compared it with the published apo-UAP56ΔN42 structure (1XTI)[31]. The folds of two N- and C-domains in URH49ΔN41 are essentially the same as the apo-UAP56ΔN42 structure[31]. However, the N- and C-domains of URH49ΔN41 were located in distinct positions from the respective domains of UAP56ΔN42 (Supplementary Fig. 10A, Table 1). The crystal of URH49ΔN41 contained SO$_4^{2-}$ and polyethylene glycol (PEG) around the interspace between N- and C-domains and the ATP-binding pocket (Supplementary Fig. 10B). This raised the possibility that the structure of apo-URH49ΔN41 containing SO$_4^{2-}$ and PEG differs from that of UAP56ΔN42 because of its interaction with these compounds.

To exclude this possibility, we generated URH49ΔN41 apo-structure models lacking SO$_4^{2-}$ and PGE by molecular dynamics analysis[32]. Among these models, the Fr48 model is the representative conformation without thermodynamical destabilization (Supplementary Fig. 10C, D). This structural model showed essentially the same structure as that of URH49ΔN41 containing SO$_4^{2-}$ and PEG (Supplementary Fig. 10A, E). Thus, we concluded that SO$_4^{2-}$ and PGE did not significantly affect the overall structure of URH49ΔN41 and continued our analysis of this model structure as the authentic apo-URH49ΔN41.

The apo-URH49ΔN41 model exhibited three different structural features compared to the apo-UAP56ΔN42 structure (Fig. 5A). First, although the amino acid sequence of the linker is the same in both helicases, the linker of UAP56 did not have any secondary structure, while this region of URH49 showed a clearly oriented α-helical structure. This structural difference in the linker part was also implicated by the finding that the R2 fragment containing the linker and C domain was derived exclusively from URH49ΔN41 but not from UAP56ΔN42 (Fig. 4E). Second, the relative orientation of the N- and C-domains differed between the apo-UAP56ΔN42 and the apo-URH49ΔN41. The overall folds of N- or C-domains were similar to each other, whose characters are conserved in the DEAD-box families[31]. (Fig. 5B, Supplementary Fig. 10E). The distinct spatial arrangement of the N- and C-domains in DEAD-box family proteins affects complex formation[20], implying that these differences may contribute to their unique apo-complex formation.

We investigated the reasons for the differential spatial positioning of the N- and C-domains between UAP56 and URH49 (Supplementary Fig. 10F). To identify the residues that underwent global structural differences between UAP56 and URH49, the alteration of all backbone dihedral angles between the two molecules were exhaustively calculated (Fig. 5C, Supplementary Table 3). Several residues were located at the linker region of UAP56 (254E–257L) and URH49 (253E–256L) and at the UAP56 243-245 loop and URH49 242-244 loop (Fig. 5D). In addition, UAP56 M243 and URH49 M242 on each loop were directly interacted with UAP56-V224 and URH49-C223, respectively (Fig. 5D). These interactions presumably altered the spatial arrangement of UAP56 M243 and URH49 M242, which in turn altered the arrangement of UAP56 D245 and URH49 D244. Then, these spatial arrangements influenced subsequent loops (UAP56 243-245 and URH49 242-244) and interdomain linkers. Moreover, the loop structures in the C-domain formed by residues 344-354 of UAP56 and residues 343-353 of URH49 (hereafter referred to as C-domain loop) are positioned differently from each other because some amino acids in this region also exhibited differential spatial positioning (Fig. 5E, Supplementary Table 3). Although the amino acid sequences constituting the C-domain loop are identical in UAP56 and URH49 (Supplementary Fig. 4), the C-domain loop of URH49 covers its own ATP-binding pocket. Currently, a possibility that the differences between UAP56-V224 and URH49-C223 affect their differential structure of interdomain linkers and C-domain loops and that URH49-C223 and UAP56-V224 may be the key amino acids responsible for their structural differences has been raised. Besides, the clarification of the precise mechanism underlying the structural differences between the two helicases awaits further structural and biochemical studies.

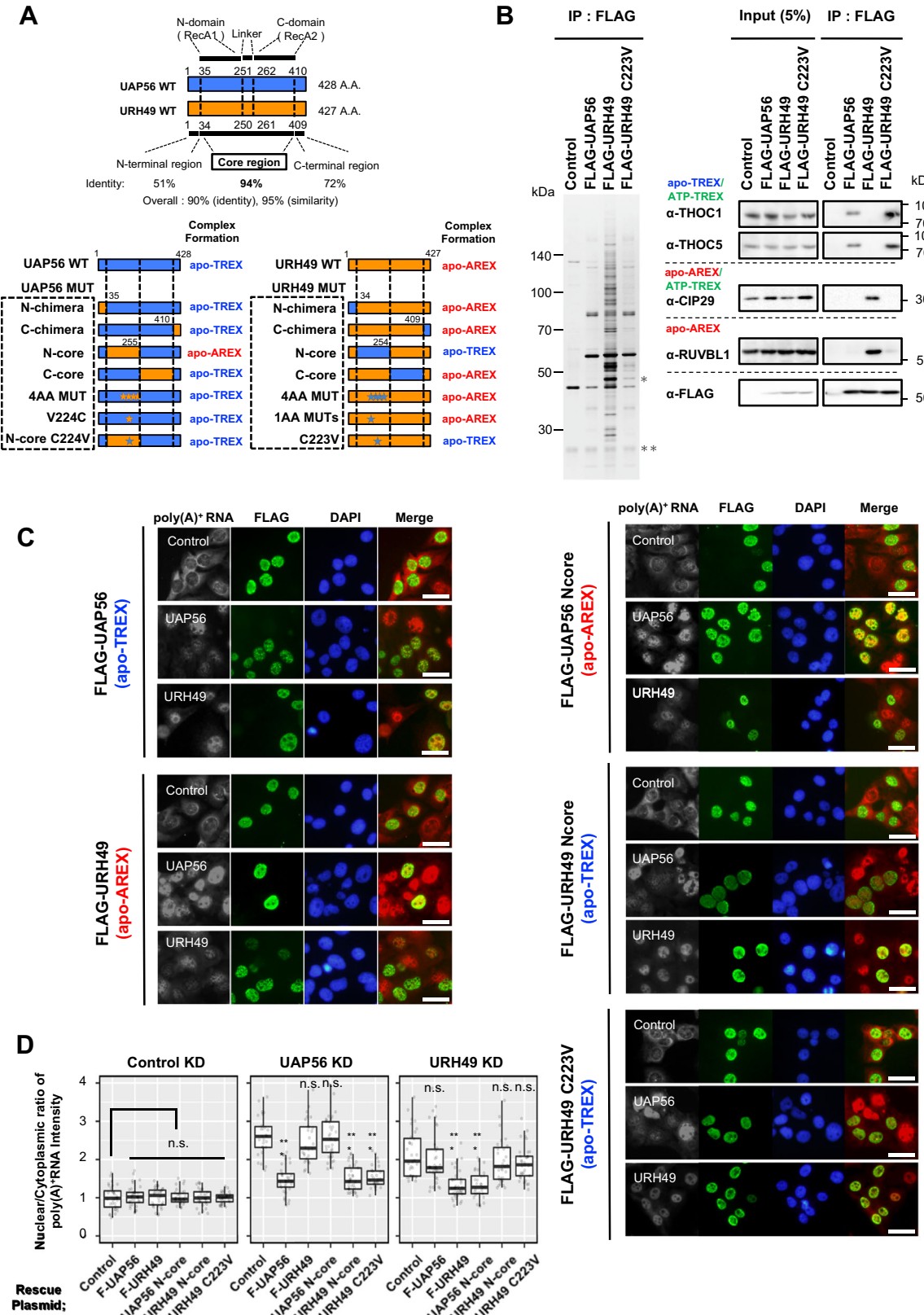

## Evolutionary diversified apo-structures from Sub2 to UAP56 and URH49

We then investigated whether the structural features of either UAP56 or URH49 observed in their apo states were conserved in yeast Sub2, the ancestor gene of UAP56. First, we performed the limited proteolysis of Sub2ΔN59, a mutant corresponding to UAP56ΔN42, in ATP-

depleted conditions. The digestion pattern of Sub2ΔN59 was similar to that of UAP56ΔN42 (Supplementary Fig. 10G). Second, we compared the structural difference of C-domains in UAP56, URH49, and Sub2 extracted from the co-crystal structure of Sub2-THO[33]. The location of the C-domain loop of Sub2 was similar to that of UAP56 (Fig. 5F). These data suggested that the structure of UAP56 is evolutionarily conserved

**Fig. 3 | A single amino acid alteration between UAP56 and URH49 impacts their apo-complex formation and specific functions. A** Diagram of amino acids homology between UAP56 and URH49 and a list of chimeric mutants analyzed in this study. **B** FLAG-URH49 C223V mutant forms the apo-TREX-like complex. Immunoprecipitation was performed using anti-DYKDDDDK tag antibody beads and Flp-In T-REx 293 cells stably expressing FLAG-tagged proteins. The precipitated sample was separated and detected by silver staining (left) or immunoblotting with the indicated antibodies (right). Single and double asterisks represented pre-cipitated FLAG-UAP56 and -URH49, and IgG light chain, respectively.
**C** Overexpression of chimeric mutants of FLAG-UAP56 or URH49 rescued the nuclear poly(A)$^+$ RNA accumulation due to UAP56 or URH49 depletion. Poly(A)$^+$ RNA (red), exogenously expressed FLAG-UAP56 or -URH49 (green), and chromo-somal DNA (blue) were visualized in U2OS cells. Scale bar, 40 μm. **D** Quantification

of the nuclear poly(A)$^+$ RNA accumulation caused by each condition in (**C**). The fold changes in the ratio of nuclear to cytoplasmic distribution of poly(A)$^+$ RNA are shown. These data were normalized to the score of control plasmid overexpression under the control knockdown condition. The signal intensities of bulk poly(A)$^+$ RNA in the nucleus and the cytoplasm were quantified from cells for each condition using ImageJ (Control knockdown: $n = 32, 33, 32, 31, 32$ and 33 cells, UAP56 knockdown: $n = 29, 33, 29, 34, 33$ and 28 cells, URH49 knockdown: $n = 32, 38, 35, 29, 31$ and 33 cells, from left to right, respectively). Boxes show the median (centerline) and upper and lower quartiles. Whiskers show the lowest and highest values. Statistical analysis was performed using the Kruskal-Wallis test followed by the Steel test. ***$p < 0.001$. n.s.: not significant. For panels (**B**–**D**), similar results were obtained in at least three independent experimental settings. Source data are provided with details of statistical tests and exact $p$-values as a Source data file.

with Sub2, while URH49 has diversified from UAP56 during evolution to form a different apo-structure.

## Discussion

In this study, we uncover unknown apo-AREX components and the molecular basis for their distinct complex formation, which is crucial for the functional divergence of both helicases playing distinct roles in mRNA processing and export.

### The apo-AREX complexes regulate gene expression of URH49 target genes

With the exception of CIP29, the details of the apo-AREX composition were not determined. Here, we identified RUVBL1, RUVBL2, ILF2, ILF3, and HNRNPM as factors that interact with URH49 by two-step affinity purification based on the apo-AREX complex under the ATP depletion condition. The RNase-treated nuclear extracts were used for immu-noprecipitation. Thus, URH49 and the AREX component identified in this study can interact even in the absence of RNA. Depletion of each apo-AREX component induced the accumulation of poly(A)$^+$ RNA in nuclear speckles. mRNAs with retained introns are tethered in nuclear speckles and thus inefficiently exported to the cytoplasm[26,27]. This observation has led to the idea that the apo-AREX complex may have a link to upstream mRNA processing, such as splicing, as well as the apo-TREX complex does[5,34].

All of the identified apo-AREX components in this study have other previously described roles in nuclear RNA dynamics. RUVBL1 and RUVBL2 have been reported to function as members of several protein complexes. They form heterodimers and function in chromatin remodeling as INO80 and SRCAP complexes[22]. As TIP160 complex components, they are also involved in the regulation of transcription via histone acetylation at promoters[22]. In addition, they are a part of the R2TP complex, which is thought to be involved in the assembly of multimeric complexes[35]. HNRNPM, which belongs to the hnRNPs family, together with various interacting factors, contributes to many aspects of RNA metabolism[23,36,37]. ILF2 and ILF3 also form heterodimers and function in RNA splicing as the LASR complex with numerous proteins, including HNRNPM[24,37]. Since other complex components were not present in the URH49 precipitate, this suggests that apo-AREX exists as a separate complex from the above complexes. In addition to the fact that many of the above complexes are involved in transcription and RNA processing, CIP29 contains an evolutionarily conserved DNA-binding motif, SAF domain, and binds to DNA, which led to the speculation that CIP29 functions in transcription[38,39]. These findings suggested that the factors identified in this study may form multiple complexes, including the apo-AREX complex and are widely involved in RNA metabolism from chromatin regulation to splicing.

Based on these insights, we expected the following potential roles of UAP56 and URH49 in mRNA processing and export through complex formation. UAP56 is involved in the splicing process by controlling spliceosome assembly through its ATPase and helicase

activities[9,10], while whether URH49 is required for these activities remains unclear. Considering this, we propose the following model. UAP56 and URH49 function in the recognition and export of target mRNAs by forming their respective complexes.

Prior to splicing, UAP56 and URH49 are associated with their apo-complexes. The DEAD-box helicases to which UAP56 and URH49 belong generally bind to RNA independently of the RNA sequence[20]. Chimeric mutants with swapped complex formation abilities exhibited a swap in target specificity between UAP56 and URH49, raising the possibility that their apo-complex components other than UAP56 and URH49 specify the selective regulation of RNA binding. Based on these results, it is likely that each apo-complex binds to specific RNA sequences or is guided by upstream processes (such as chromatin regulation), leading to subsequent interactions with the target mRNA by UAP56 and URH49. Subsequently, through the remodeling of each apo-complex into the ATP-TREX complex, the ATPase and helicase activities of UAP56 and URH49 are activated, leading to the splicing of the respective target pre-mRNA and, ultimately, mRNA export. Further studies will uncover how two closely related complexes recognize their target transcripts, which will reveal the individual functions of these two complexes in coupling the processes from mRNA transcription to export.

### The apo-AREX complexes are implicated in cell proliferation and cancer progression

UAP56 is continuously expressed during the cell cycle, while URH49 is expressed specifically during the proliferation phase and not during the quiescent phase[14]. URH49 is required for gene expression of sub-sets of key regulators of mitosis[15] and cytokinesis[11]. The apo-AREX components were also required for the expression of representative targets regulated by URH49. These results indicate that the apo-AREX complex is implicated in cell proliferation via regulation of its target gene expression.

Aberrant expression of UAP56 and URH49 is involved in tumorigenesis and cancer progression. UAP56 is upregulated in colorectal and ovarian cancers and is associated with their progressions[40,41]. The association between URH49 and cancer has been reported and observed more often than that of UAP56. In Cancer Genome Atlas (TCGA), pan-cancer cohort analysis showed that URH49 is upregulated in 18 cancer types than normal tissue[42]. Actually, aberrant upregulation of URH49 is observed in various cancer tissues and cancer cell lines and is positively correlated with advanced clinical stage and poor prognosis[42–44]. URH49 is important for the gene expression involved in cell proliferation and promotes malignancy of these cancer[42,44]. CIP29 is highly expressed in various cancers and is associated with cancer malignancy[7,45]. Overexpression of RUVBL1, RUVBL2, ILF2, ILF3, and HNRNPM are observed in various cancers with their progression (RUVBL1 and RUVBL2[22,46], ILF2 and ILF3[47–49], HNRNPM[50,51]). Thus, it is possible that the apo-AREX components are important regulators of gene expression in cancer.

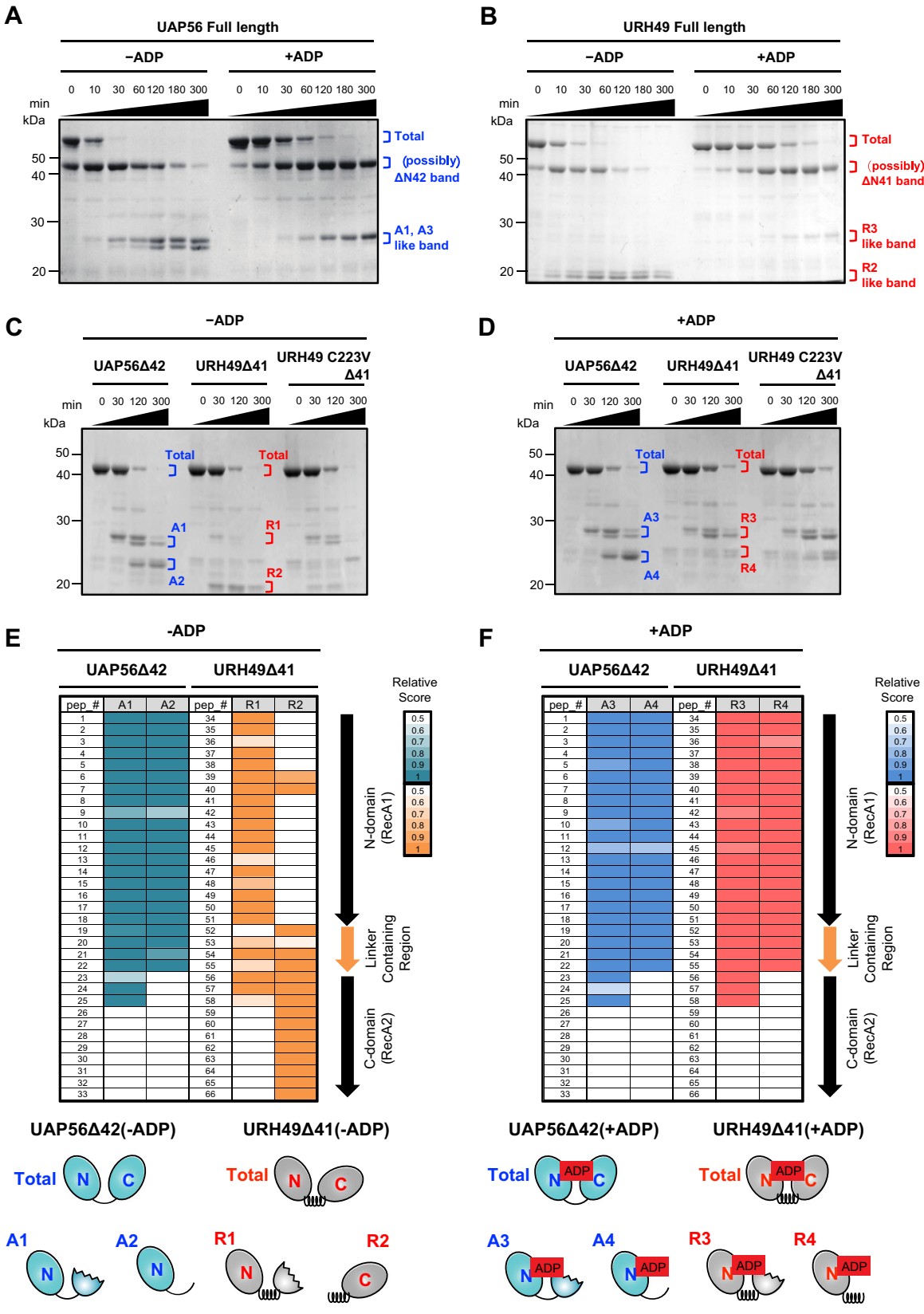

Therefore, agents that impair the apo-AREX complex expressions and/or activity could be potential targets for cancer therapy. Indeed, YM155, an inhibitor of ILF3, and CB-6644, an inhibitor of RUVBL1 and RUVBL2 subcomplex, exhibit anticancer activities[52,53]. The research focusing on the regulation of the apo-AREX complex may contribute to a therapeutic benefit in cancer.

## Complex formation and structure of UAP56 and URH49

We observed that UAP56 and URH49 have different structural features in the apo-state and are remodeled to similar structures upon the ADP binding. The crystal structure of UAP56 and URH49 in apo-state showed the distinct configuration of the N- and C-domain, probably due to different linker structures. UAP56 interacts with the THO

**Fig. 4 | UAP56 and URH49 have different apo-structural features but similar ATP-binding structural features, which were correlated with their complex formation.** Each purified protein was treated with trypsin. Aliquots were taken at each time point, separated by SDS-PAGE, and detected by Coomassie staining. **A**, **B** Full-length UAP56 and full-length URH49 had different partial digestion patterns in the absence of ADP but similar limited proteolysis patterns upon ADP addition. **C** UAP56Δ42 and URH49Δ41 showed different limited proteolysis patterns, and URH49 C223VΔ41 showed a pattern similar to apo-UAP56Δ42 but not apo-URH49Δ41 in the absence of ADP. **D** UAP56Δ42 and URH49Δ41 showed similar limited proteolysis patterns upon ADP addition. **E**, **F** Top: Analysis of cleavage sites

by limited proteolysis. The "A1-4", "R1-4", and "Total" products obtained by limited proteolysis of UAP56Δ42 and URH49Δ41 in (**C**, **D**) were analyzed by LC-MS/MS. Relative peptide scores were obtained by dividing the detected prot-score of each peptide fragment derived from "A1-4" and "R1-4" by "Total". The start site, the end site, and the relative score of each peptide were described in Supplementary Table 2. Bottom: Limited digestion models of UAP56 (blue) or URH49 (gray) predicted from the peptide containing within "A1-4" or "R1-4". For panels (**A**–**D**), similar results were obtained in at least three independent experimental settings. Source data are provided as a Source Data file.

subcomplex via their N- and C-domains within the reconstituted UAP56-THO subcomplex[54]. These interactions were also conserved in the crystal structure of yeast Sub2-Tho[33,55]. These findings suggest that the spatial arrangement of N- and C-domains of UAP56 is important for the formation of the apo- and ATP-TREX (UAP56) complex. The limited proteolysis of purified UAP56 and URH49 in solution suggested that a significant portion of the apo-structures of UAP56 and URH49 may differ in solution. Additionally, the limited digestion products of the URH49-C223 mutant, which exhibited a switch in complex formation from the apo-AREX complex to the apo-TREX complex, showed a pattern similar to that of UAP56 under apo-condition. Therefore, the difference in the configurations between UAP56 and URH49 structures

## Table 1 | Data collection and refinement statistics

| | URH49Δ41 |
|---|---|
| Data collection | |
| Source | BL26B1, SPring-8 |
| Wavelength (Å) | 0.90 |
| Detector | CCD (Rayonix MX225) |
| Space group | $P2_1$ |
| Unit cell (Å, °) | $a = 34.97$, $b = 96.61$, $c = 59.76$ $\beta = 95.51$ |
| Resolution limit (Å) | 50–1.82 (1.93–1.82) |
| Total reflections | 165397 (22373) |
| Unique reflections | 68537 (10621) |
| Completeness (%) | 98.5 (94.2) |
| Multiplicity | 2.4 (2.1) |
| *R*-merge | 0.046 (0.373) |
| *R*-mean | 0.058 (0.481) |
| CC(1/2) | 0.998 (0.843) |
| Mean *I*/σ(*I*) | 12.4 (2.09) |
| Structure determination | MR with 1XTI |
| Refinement | |
| Program used | *phenix.refine* |
| Resolution range (Å) | 37.5–1.82 (1.88–1.82) |
| No. of reflections used | 34958 (2623) |
| Completeness (%) | 99.1 (91.0) |
| Residues | 376 (resid. 13-388) |
| PO4/PGE/EDO/HOH | 2/1/19/177 |
| Wilson *B*-factor (Å²) | 26.1 |
| Bond-length r.m.s. (Å) | 0.007 |
| Bond-angle r.m.s. (°) | 0.86 |
| Ramachandran outlier (%) | 0.0 |
| Crash score | 7.35 |
| *R*<sub>work</sub> | 0.186 (0.263) |
| *R*<sub>free</sub> | 0.222 (0.312) |

Values in the parentheses are the most higher resolution shell.
$R_{merge} = \sum_{hkl}\sum_i |I_i(hkl) - \langle I(hkl)\rangle|/\sum_{hkl}\sum_i I_i(hkl)$, where $I_i(hkl)$ is the $i$th observation of the reflection $(hkl)$ and $\langle I(hkl)\rangle$ is the mean intensity of the $(hkl)$ reflection.

may play a pivotal role in their unique complex formation. However, it's worth noting that the crystal structure generally reflects one of the possible structures in solution, and the extent to which the two domains differ in solution has not been thoroughly examined. In addition, there may be unidentified factors other than the observed differences in the crystal structures of UAP56 and URH49 that influence complex formation. One successful way to solve this issue is to generate the apo-AREX, apo-TREX, ATP-TREX (UAP56), and ATP-TREX (URH49) complexes and compare their structures at high resolution using cryo-electron microscopy.

The loop structure within the C-domain of apo-URH49ΔN41 covers the ATP-binding pocket in their N-domain. While a more precise understanding of the differences between UAP56 and URH49 requires their ATP-binding affinity analysis, our results imply that the URH49 C-domain loop likely gives URH49 less affinity for ATP in the apo-state, allowing it to maintain a different conformation from the apo-state of UAP56. Although several DEAD-box proteins inhibit their ATP binding by intramolecular interactions[20], the arrangement of the loop in the C-domain is not observed among other DEAD-box proteins, suggesting that URH49 has evolutionarily acquired a unique structure to repress ATP binding. Consistent with our expectation, Sub2 exhibited sufficient helicase activity as well as UAP56[56]. In addition, CIP29 stimulates the ATPase activity of URH49, followed by ATP binding in URH49[57], implying that URH49 forms the apo-AREX complex as a steady state, then remodels to the ATP-TREX complex in the cell.

Orthologs of UAP56 and URH49 are present in vertebrates, while only the ortholog of UAP56 is present in insects, implying that the ortholog of URH49 diversified during the evolution between vertebrates and invertebrates. Furthermore, amino acids corresponding to human UAP56-V224 and URH49-C223 are already present in orthologs of UAP56 and URH49. Thus, the amino acid substitution between UAP56-V224 and URH49-C223 probably occurred after diversification.

Integrating our findings into previous observations, we proposed the following model that UAP56 and URH49 form the apo-TREX and -AREX complex based on their apo-conformation (Fig. 6). The binding of ATP into UAP56 and URH49 promotes the conformational change to a highly similar closed conformation, triggering the remodeling of the respective apo-complex to the ATP-TREX complex. It has been implicated that UAP56 and URH49 function in selective mRNA export by forming respective complex formation. Therefore, we provided the possibility that diversified apo-structures of UAP56 and URH49 derived from Sub2 have contributed to the organization of gene regulation in humans. Further progress in genome analysis of other species is expected to advance our understanding of the diversification of UAP56 and URH49.

Interestingly, amino acids in UAP56 and URH49 required for selective circular RNA export are not linked to the apo-complex formations. ATPase activities of UAP56 and URH49 are not required for circular RNA export[29]. While ATP loading and ATPase activity are known to be essential for their function of mRNA export[11,21]. In addition, there is no significant difference in length between the mRNAs selectively exported by the two helicases and these pre-mRNAs[11]. Thus, the underlying mechanism of circular RNA export needs to be

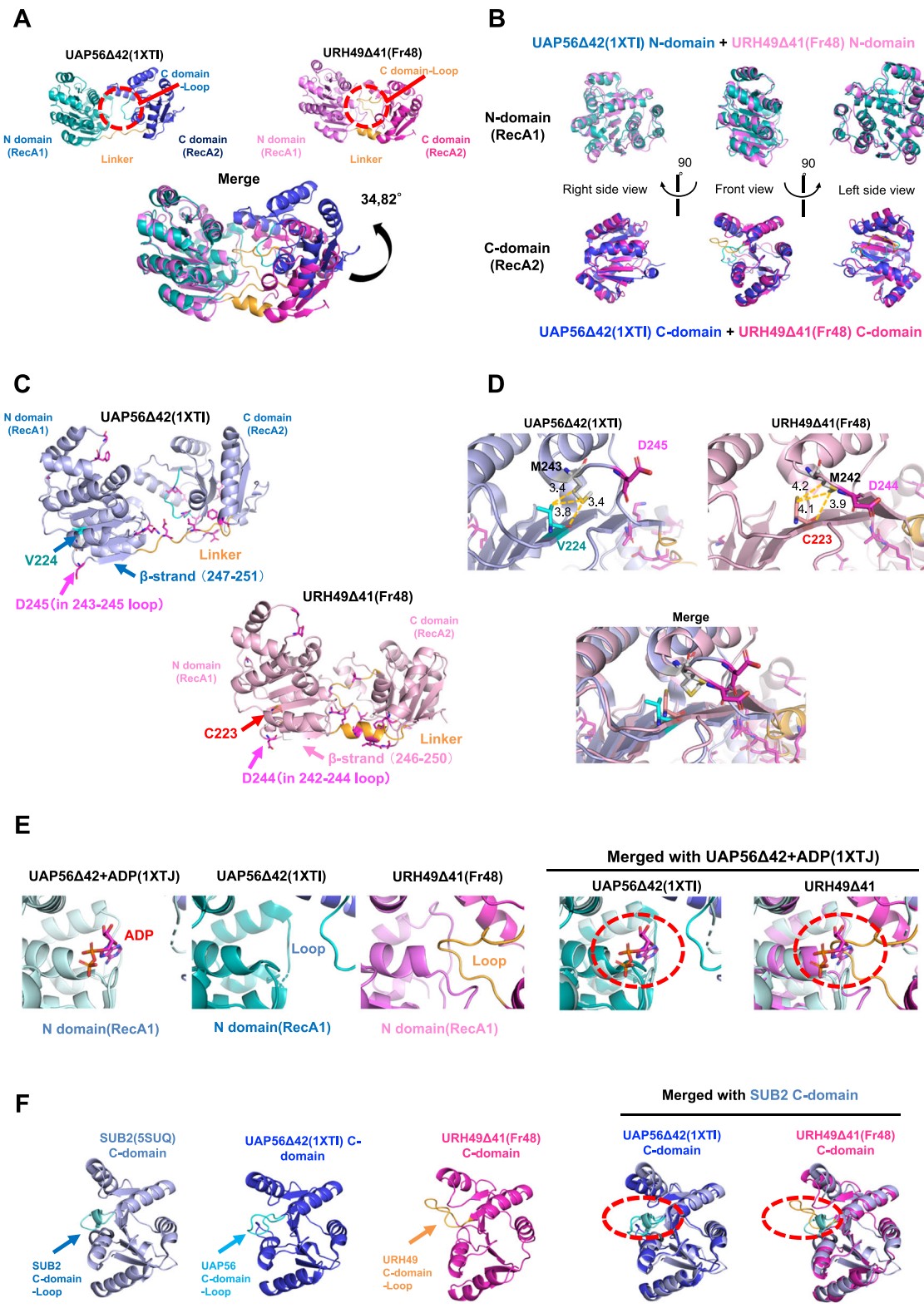

**Fig. 5 | Structural comparison between UAP56, URH49, and Sub2. A** Comparison of the structure of apo-UAP56Δ42 (PDB ID: 1XTI) and the structural model of apo-URH49Δ41 (Fr48) which is generated by molecular dynamics analysis of the structure of URH49Δ41 (8IJU). Detail of the generation of the Fr48 model were described in Supplementary Fig. 10C, D. By aligning the N-domain of both structural models using pyMOL, the difference in the angle of the C-domains was calculated. **B** Comparison of N-domain and C-domain between apo-UAP56Δ42 crystal (1XTI) and apo-URH49Δ41 model structure (Fr48). **C** The amino acid residues that may contribute to the differences in the global structures of UAP56 (1XTI) and

URH49 (Fr48). These residues highlighted in magenta were located in the linker region and C-domain loop. **D** Direct interactions between V224 of UAP56 and M243 of UAP56, C223 of URH49, and M242 of URH49, respectively. The distance between each side chain was calculated using pyMOL. **E** Top: the C-loop of the apo-URH49Δ41 structural model (Fr48) was located as covering the ATP-binding pocket of apo-URH49Δ41. Bottom: the structure of ADP-UAP56Δ42 (1XTJ) was overlaid to the structure of apo-UAP56Δ42 (1XTI) or the structural model of apo-URH49Δ41 (Fr48). **F** Loop structure in the C-domain of the apo-Sub2Δ59 (5SUQ) was overlaid to apo-UAP56Δ42 (1XTI) or URH49Δ41 structural model (Fr48).

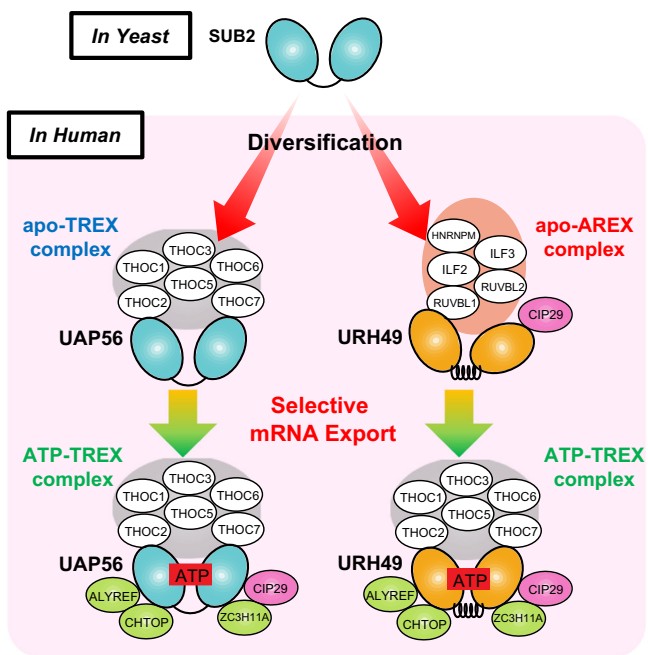

**Fig. 6 | Diversified structures and mRNA export machineries.** The model of selective mRNA transcription and export machinery is driven by structural diversification from yeast Sub2 to human UAP56 and URH49.

investigated separately from that of mRNA export, including whether the complex formation is required for circular RNA export.

During evolution, many RNA-binding proteins have functionally diversified to execute well-tuned gene expression contributing to the complexity of living organisms. These include UAP56 and URH49, which have diversified to form different complexes and function in selective mRNA processing and export[15]. In addition to both helicases, several key mRNA processing and export factors such as NXFs, NXTs, DDX19s, and SR proteins have evolutionarily diversified from yeast to human. Some factors have gained different target specificities from their originated paralogs, but the molecular mechanism behind these differences is mostly unknown[58]. Further elucidation of the diversification of mRNA export-related proteins will uncover the mechanistic insight into the accurate gene expression through mRNA processing and export in humans and how it has developed during evolution.

## Methods

### Cell culture and establishment of stable cell line
U2OS was obtained from ATCC (HTB-96), MCF7, and A549 cells were obtained from JCRB (JCRB0134, JCRB0076), Flp-In T-REx 293 cell was obtained from Thermo Fisher Scientific (Waltham, MA). These cells were maintained in Dulbecco's Modified Eagle's Medium (Fujifile Wako, Tokyo, Japan) supplemented with 10% heat-inactivated fetal bovine serum at 37 °C. Flp-In T-REx 293 cells stably expressing 3x FLAG-tagged protein were obtained by the transfection of pcDNA5 3x FLAG-tagged protein expression vector with pOG44, respectively.

### Reagents and antibodies, preparation of serum
4′, 6-diamidino-2-phenylindole (DAPI) was purchased from Fujifilm Wako. Antibodies were obtained as follows: FLAG M2 mouse monoclonal antibody (1:3000 dilution; F1804, Sigma-Aldrich Japan, Tokyo, Japan), rabbit anti-β-actin antibody (1:3000 dilution; A2066, Sigma-Aldrich Japan), rabbit anti-HNRNPM antibody (1:2000 dilution; HPA024344, Sigma-Aldrich Japan) and mouse anti-SRRM2 antibody (1:2000 dilution; S4045, Sigma-Aldrich Japan), HA (12CA5) mouse monoclonal antibody (1:2000 dilution; GTX16918, GeneTex, Irvine, CA), mouse anti-GAPDH antibody (1:2000 dilution; 016-25523, Fujifilm

Wako), sera against THOC1 (1:1000 dilution), THOC2 (1:1000 dilution), THOC5 (1:1000 dilution), ALYREF (1:1000 dilution), CIP29 (1:1000 dilution), UAP56 (1:1000 dilution) and URH49 (1:1000 dilution) have been described previously[11]. Anti-RUVBL1 (1:1000 dilution), anti-RUVBL2 (1:1000 dilution), anti-ILF2 (1:1000 dilution), and anti-ILF3 sera (1:1000 dilution) were prepared from immunized Wistar female rats as described previously[59] in accordance with the recommendations in the Guide for the Care and Use of Laboratory Animals of the Animal Committee in Kyoto University (Animal experiments were approved by the Committee on the Ethics of Animal Experiments of Kyoto University, Experiment permission number: Lif-K17002). The antibodies used in this study are listed in Supplementary Table 4.

### Plasmids, primers and siRNAs
To construct the following plasmids, the fragments were obtained by PCR amplification with the addition of restriction enzyme sites or infusion enzyme sites at both ends. pcDNA5-3xFLAG and pcDNA5-HA vectors were generated as described previously[11,60]. 3×FLAG-UAP56, 3×FLAG-URH49, 3×FLAG-CIP29, 3×FLAG-RUVBL1, 3×FLAG-RUVBL2, 3×FLAG-ILF2, 3×FLAG-ILF3, and 3×FLAG-HNRNPM expression vectors were generated by the insertion of the respective open reading frame into pcDNA5-3×FLAG, respectively. The HA-CIP29 expression vector was generated by the insertion of the CIP29 open reading frame into pcDNA5-HA. To construct GST-UAP56 and GST-URH49, and their derivative mutant expression plasmids, respective open reading frames of UAP56, URH49 and their mutants were inserted into pGEX6p2. The GST-Sub2Δ59 (60–446 amino acids) expression plasmid was constructed by inserting the respective region into pGEX6p2. MBP-RUVBL1 (250–456 amino acids), MBP-RUVBL2 (1–225 amino acids), MBP-ILF2 (240–390 amino acids), and MBP-ILF3 (280–355 amino acids) expression plasmids were constructed by inserting their respective region into pMALc2X. To construct mutants of FLAG-UAP56 or -URH49 expression plasmids, overlap extension PCR was performed to induce the mutation. The construction of the plasmids was confirmed by sequencing. The primers and siRNAs used in this study are listed in Supplementary Tables 5 and 6.

### Plasmid or siRNA transfection
Transient transfection of siRNA and plasmids was performed using Lipofectamine 2000 (Thermo Fisher Scientific) according to the manufacturer's instructions.

### Total, cytoplasmic and nuclear RNA isolation
Total RNA was isolated by Sepasol-RNA I super G (Nacalai Tesque, Kyoto, Japan) according to the manufacturer's instructions. For cytoplasmic RNA preparation, the cells were treated with lysis buffer (20 mM Tris-HCl (pH 8.0), 200 mM NaCl, 1 mM $MgCl_2$, 1% NP-40) on ice for 5 min. The cytoplasmic fraction was isolated by brief spin while the nuclear fraction was prepared from the precipitate.

### Quantitative and semi-quantitative reverse transcription-polymerase chain reaction (RT-PCR)
Quantitative RT-PCR (RT-qPCR) was performed with TB Green Premix Ex Taq II (TakaraBio, Tokyo, Japan) and analyzed by Thermal Cycler Dice real time system II (TakaraBio). PGK1 was used for standardization. The quantity of each mRNA was calculated by threshold cycle (Ct) values. The relative expression of each mRNA was evaluated by the values of $2^{[Ct (TBP) - Ct (each mRNA)]}$. Primer sets are listed in Supplementary Table 6.

### Immunoprecipitation, immunoblotting, LC-MS/MS analysis and silver staining
Preparation of nuclear extract and immunoprecipitation were performed as described previously[11]. Briefly, nuclear extract was incubated for 30 min at 20 °C for the depletion of ATP and centrifuged to

recover the supernatant. Then, RNaseA (100 ng/μL), ATP (500 μM), MgCl$_2$ (3.2 mM) and creatine phosphate (20 mM) were added to the nuclear extract, and the reaction mixture was incubated for 30 min at 30 °C. In the ATP (−) condition, ATP, MgCl$_2$, and creatine phosphate were omitted. After a brief spin, the clear supernatant was mixed with anti DYKDDDDK tag antibody beads (Fujifilm Wako) or anti HA antibody beads (Fujifilm Wako) and rotated overnight at 4 °C. The beads were extensively washed with PBS containing 0.1% TritonX100, 0.2 mM PMSF and 0.5 mM DTT to remove nonspecifically bound proteins. The proteins attached to the beads were dissolved in SDS sample buffer (250 mM Tris-HCl, 1% sodium lauryl sulfate), 0.002% bromophenol blue and 40% Glycerol for 10 min at 37 °C. The eluate was recovered in a new tube and DTT was added to 10 mM and boiled for 2 min. For tandem immunoprecipitation, the proteins attached to the beads were eluted with FLAG peptide (M&S TechnoSystems Inc, Osaka, Japan) or HA peptide (MBL, Nagoya, Japan).

Samples were separated by SDS-polyacrylamide gel electrophoresis (SDS-PAGE) and blotted onto polyvinylidene difluoride membrane (Pall, Ann Arbor, MI). The blotted membrane was blocked with PBS containing 0.1% polyoxyethylene sorbitan monolaurate (Tween20) and 5% skim milk for 1 h and reacted with primary antibodies at 4 °C overnight with gentle rotation. The membrane was extensively washed with PBS containing 0.1% Tween20. Secondary antibody conjugated with horseradish peroxidase was reacted with the membrane by rotating for 2 h. After extensive washing, the membrane was reacted with a chemiluminescence reagent (Millipore, Darmstadt, Germany). Signals were detected with LAS 4000 mini (GE Healthcare Japan, Tokyo, Japan).

The LC-MS/MS analysis was performed by Q Exactive Plus (Thermo Fisher Scientific, Waltham, MA). As outputs of the LC-MS/MS analysis, prot score was calculated using Mascot software (Matrix Science, London, UK). For factors with different prot_acc but the same GeneName, only the largest prot_score is listed. For comparison, protein scores were calculated by subtracting the prot_score of the control from each data. Gene ontology (GO) was analyzed using Database for Annotation, Visualization and Integrated Discovery (DAVID: version 6.7)[61].

For silver staining, proteins were separated with SuperSep™ Ace, 5%–20%, 17-well (Fujifile Wako). Silver staining was performed as described previously[62].

## Immunofluorescence staining
Cells (5 × 10$^4$ cells/mL) on glass coverslips in a 12-well plate were cultured for 24 h and transfected with siRNA or plasmid. After a 48 h incubation, cells were fixed in 4% formaldehyde in PBS, permeabilized with 0.1% Triton X-100 in PBS, and blocked with 6% bovine serum albumin (BSA) in PBS. The coverslips were reacted with primary antibodies in 2% BSA in PBS, secondary antibody conjugated with Alexa-488 or Alexa-594 (Molecular Probes, Eugene, OR) and DAPI to counterstain the nuclei. Fluorescence images were obtained with a fluorescent microscopy, Axioplan 2 (Carl Zeiss, Germany) or FV10i (Olympus, Tokyo, Japan), a laser scanning confocal microscopy, using the ×60 objective lens. Line Plot analysis was performed using FV10-ASW v4.1 software (Olympus).

## RNA-fluorescence in situ hybridization (FISH)
RNA-FISH was performed as described previously[11]. Briefly, cells (5 × 10$^4$ cells/mL) were inoculated on glass coverslips in a 12-well plate, cultured for 24 h and transfected with siRNA or plasmid. After 24 to 48 h incubation, cells were fixed with 10% formaldehyde in PBS for 20 min and permeabilized in 0.1% Triton X-100 in PBS for 10 min. The coverslip was washed three times with PBS for 10 min and once with 2× Standard Saline Citrate (SSC) for 5 min. Cells were prehybridized with ULTRAhyb-Oligo Hybridization Buffer (Ambion, Austin, TX) for 1 h at

42 °C in a humidified chamber. Then, they were treated with 10 pmol Alexa Fluor 594-labeled oligo-dT$_{45}$ probe (Molecular Probes) overnight. Cells were washed for 20 min at 42 °C with 2 × SSC, 0.5 × SSC, and 0.1 × SSC. Nuclei were counterstained with DAPI. Fluorescent images were obtained with Axioplan 2. Poly(A)$^+$ RNA signal intensities in the nucleus and the cell were calculated with ImageJ software (https://imagej.nih.gov/ij/).

## Protein expression and purification
GST-fusion proteins were produced in *E. coli* BL21 strain. The production of recombinant protein was induced by the addition of 0.05 mM IPTG at 18 °C overnight. Cells were pelleted by centrifugation at 6000 × *g* for 10 min. The pellet was resuspended in PBS containing 0.2 mM phenyl methyl sulfonyl fluoride (PMSF) and 1 mM dithiothreitol (DTT) and then sonicated 30 s four times on ice. The clear lysate was obtained by centrifugation at 8000×*g* for 15 min and transferred to a new tube. Glutathione-fixed beads (GE Healthcare) were added to the clear lysate and rotated for 3 h at 4 °C. After the extensive washing with PBS containing 0.2 mM PMSF and 1 mM DTT, precision protease (GE Healthcare) was added to remove the GST-tag and rotated overnight at 4 °C. The eluate containing the GST-tag removed protein was further purified on a gel filtration column, HiPrep 16/60 Sephacryl S-100 HiResolution (GE Healthcare). The purity and concentration of recombinant protein were confirmed by SDS-PAGE followed by Coomassie Brilliant Blue R-250 (Nacalai Tesque) staining.

## Limited proteolysis
Recombinant protein was incubated with 1/100 (weight ratio) of trypsin (Promega Japan) at 25 °C for 0, 30, 120, or 300 min. In the ADP condition, ADP (1 mM) and MgCl$_2$ (10 mM) were added to the reaction mixture. The digestion was stopped by adding an equal volume of SDS sample buffer. Samples were boiled for 2 min, then separated by SDS-PAGE and stained with Coomassie Brilliant Blue R-250. The LC-MS/MS analysis for some of the separated bands was performed by Q Exactive Plus.

## Crystallization and crystal structural analysis
URH49ΔN41 was concentrated at 5 mg/mL and crystallized by the sitting-drop vapor diffusion. Briefly, 1 μL of a protein solution was mixed with 1 μL of a mother liquid containing 0.2 mM NaPO4, pH 8.5, 30 % (w/v) PEG3350 at 20 °C. The diffraction of the crystals was confirmed by an in-house Bruker Hi-star detector after flash-cooling in a cold nitrogen gas stream (100 K) with 25% (v/v) ethylene glycol. The diffraction images were collected at 100 K (in a cold nitrogen gas stream) on a Rayonix MX225 CCD detector (Rayonix, Evanston, IL) with a wavelength of 0.9 Å at BL26B1 in SPring-8 (Hyogo, Japan). The resulting data sets were processed, merged, and scaled using XDS (version Mar. 31, 2022)[63]. The structure was solved by molecular replacement with UAP56ΔN42 (Protein Data Bank entry 1XTI) using a search mode by Molrep implemented in CCP4i 7.0.073 software[64]. The model was refined using PHENIX 1.20.1 software[65], rebuilt using COOT 0.8.9[66] and further modified based on sigma-weighted (2|Fo|-|Fc|) and (|Fo|-|Fc|) electron density maps. Protein structure images were depicted using PyMOL software (The PyMOL Molecular Graphics System, Version 2.0 Schrödinger, LLC).

## Molecular dynamics (MD) analysis
MD simulation was performed for the apo-form of URH49ΔN41 using Desmond Molecular Dynamics System, version 5.2 (D. E. Shaw Research, New York, NY)[32]. The atomic coordinates of SO$_4^{2-}$ and PGE were removed from the crystal structure of their complex with URH49ΔN41 to generate the initial structure for the simulation. First, the structure was preprocessed with Protein Preparation Wizard of Maestro (version 11.4), the GUI for Desmond, to assign bond orders,

add hydrogens, and create disulfide bonds. Then, it was solvated in a box with a buffer distance of 10 Å to the boundary. Afterward, solvation was performed in a box with a buffer distance to the boundary of 10 Å. Sodium and chloride ions were added to neutralize the entire solvated system. OPLS_2005 force field[67] and SPC model[68] were used for the protein and water molecules, respectively. After relaxing the system according to the Maestro's default relaxation protocol, an MD run was performed in the constant-NPT ensemble at 300 K and 1.013 bar for 1 μs. The coordinates were recorded every 1 ns to yield 1001 snapshots. Otherwise, the default setting in Desmond was adopted. The resulting MD trajectory was equidistantly divided into 101 frames so that each frame could contain ten consecutive snapshots. Then the Root Mean Square Deviation (RMSD) values between main chains of arbitrary two frames were calculated to generate an RMSD matrix. In the matrix, frames with an RMSD less than 2 Å were assigned to belong to the same cluster. In each cluster, the frame with the minimal RMSDs to the other members was considered a representative structure of the cluster.

### Structural analysis based on local conformation

A comprehensive analysis of the dihedral angle changes along the backbones of both molecules was conducted to identify the residues that may be responsible for the global structural differences between UAP56 and URH49. First, all the dihedral angles were calculated from the coordinates of four consecutive atoms along each rotatable bond and classified into six conformations. Each conformation class was designated by the corresponding senary number, where ±synperiplannar (±sp) conformation (dihedral angle from −30° to 30°) corresponds to 0, +synclinal (+sc) conformation (dihedral angle from 30° to 90°) to 1, +anticlinal (+ac) conformation (dihedral angle from 90° to 150°) to 2, ±antiperiplannar (±ap) conformation (dihedral angle from 150° to 180° and from −180° to −150°) to 3, -anticlinal (-ac) conformation (dihedral angle from −150° to −90°) to 4, and -synclinal (-sc) conformation (dihedral angle from −90° to −30°) to 5. Alterations in all main-chain dihedral angles between UAP56 and URH49 were estimated as the difference between these cyclic numbers (for example, the difference between 5 and 1 was not 4 but 2). The difference in the cyclic secondary number with 1 means a transition to the neighboring conformation, the difference in the cyclic secondary number with 2 means a transition to the next-to-next conformation, and the difference in the cyclic secondary number with 3 means a transition to the most distant conformation. Because there are one or more potential energy barriers to the most distant conformation, the transition there requires significant conformational change, and thus the residues with dihedral angles whose senary number was altered by 3 were assumed to contribute to the whole structural variation between UAP56 and URH49.

### Quantification and statistical analysis

RT-qPCR results were quantified using Thermal Cycler Dice real time system II (TakaraBio). FISH data was quantified using ImageJ software. Immunofluorescence staining data were quantified using FV10-ASW v4.1 software (Olympus). LC-MS/MS data were analyzed using Mascot software (Matrix Science, London, UK). The statistical significance for two-group and multiple comparisons was tested using R software[69], as indicated in the legend of each figure. Non-adjusted (two-group comparison) and adjusted (multiple comparisons) P-values are indicated in each figure. In box plots, the first and third quartiles are indicated by both ends of the box, the median is indicated by a vertical line in the box, and the minimum and maximum excluding outliers are the ends of the whiskers. The outliers are indicated with open circles.

### Reporting summary

Further information on research design is available in the Nature Portfolio Reporting Summary linked to this article.

## Data availability

The data supporting the findings of this study are available from the corresponding authors upon request. Atomic coordinates and structure factors for the reported crystal structures (URH49Δ41) have been deposited with the Protein Data bank under accession number 8IJU. Source data are provided with this paper.

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

## Acknowledgements
We thank Mr. Yuzo Watanabe for his help with LC-MS/MS analysis and our lab members for their constructive discussions. Diffraction data were collected at the BL26B1 and BL44XU stations of SPring-8 (Hyogo, Japan) with the approval of JASRI (proposal nos. 2015A1063, 2015B2063 and 2017B6750). We thank Editage (www.editage.com) for English language proofreading. This work was supported in part by "Grants-in-Aid" from JSPS KAKENHI (Grant Numbers 26292053, 17K19232, 19K22280, 19H02884, 21K19078 and 22H02264 to S.M., 19K15807 to K.F.). This work was also supported in part by "Grants-in-Aid" from The Sasakawa Scientific Research Grant from The Japan Science Society to K.F.

## Author contributions
K.F. and S.M. conceived and designed this study; K.F. performed the experiments and analyses, organized the data and drafted the manuscript; M.Ito, M.Irie, K.H., Y.I., H.Y., and T.Y. performed biological experiments; M.K. performed MD analysis; B.M. performed crystal structure analysis; K.F., N.F., A.M., and S.M. analyzed the results and wrote the paper. All authors reviewed the final manuscript.

## Competing interests
The authors declare no competing interests.
