## [Peer Review File · Nature Communications]

Structural differences between the closely related RNA helicases, UAP56 and URH49, fashion distinct functional apo-complexesREVIEWER COMMENTS

Reviewer #1 (Remarks to the Author):

In this manuscript, using tandem-immunoprecipitation and mass spectrometry, Fujita et al. identified several novel key components together with URH49, a paralogue of helicase UAP56 in mammals, to form the apo-AREX complex. Although apo-AREX and apo-TREX complexes consist of distinct different components, they rebuilt into same ATP-complexes. The authors found that UAP56 and URH49 play key roles in dictating the formation of these two distinct apo-complexes. Surprisingly, only a single amino acid substitution in UAP56 or URH49 N-domain altered the formation and function of the apo-complexes. Further, using limited proteolysis assay, the authors claimed that UAP56 and URH49 exhibited different apo-structures, but with similar ADP binding conformation. They also solved the crystal structure of apo-URH49 Δ N41, which showed three different features compared to the apo-UAP56 Δ N42 structure. Based on these observations the authors proposed that UAP56 and URH49 form different complexes in the absence of ATP based on their diverse apo-conformation. This manuscript present mechanistic study that is important for understanding selective nuclear export of RNAs. The data are solid and the manuscript is overall clearly written. I have a couple of comments that the authors could consider to improve the study.

1. The authors performed immunoprecipitation with Flag-URH49 in 293 cells and detected many candidates for the apo-AREX components. Are the interactions of these new components with URH49 mediated by RNA? What is the function of the apo-AREX complex formed in the absence of ATP (or ADP). Considering that mRNA export requires ATP, it does not seem like it is important for this process. As this is the major finding of the work, the authors could discuss more, especially on the potential roles of this complex.
2. FISH assay was performed to detected the nuclear poly(A) RNA accumulation in apo-AREX components depleted U2OS cells. What is the reason that the authors selected the U2OS cells in the FISH assay? They performed the immunoprecipitation in the 293 cells, dose the FISH assay can get the same result as that in the U2OS cells?
3. The digested fragments of URH49 Δ N41 and full length seems obtain different patterns, which indicates the depleted N-terminal region may change the behaviour of URH49 against trypsin digestion. The R1 and R4 bands are very week, and the A1, A3 R3 include two bands, maybe the limited proteolysis assay should be further optimized, such as incubating with less enzyme or shorter time. Besides, the limited proteolysis assay is not the best way to detect the protein conformation in solution, more evidence should be provided to confirm this observation.
4. In Extended Data Fig.7A, the authors showed the URH49 Δ N41(-ADP) was digested into two separate fragments R1 and R2, which was different from the other three samples. There seems no solid evidence to support this model.
5. The two RecA domains within UAP56 do not adopt fixed relative orientation in solution, and the crystal structure reported here reflects just one state of URH49 in solution. So, more evidences are required to support the conclusion that the apo-structures of UAP56 and URH49 make them to integrate into separate complexes.
6. It is intriguing that no structural difference was observed with the key amino acid (URH49 C223 and UAP56 V224) that determines the different apo-complex of UAP56 and URH49. The author proposed that the difference in the loop structure might be caused by this amino acid. It would be great if they could use molecular modelling to examine this possibility.
7. It would be important for the authors to at least discuss how the difference in their apo-complexes determines the substrate specificity of UAP56 and URH49.
8. In Extended Data Fig.7C, I guess the sample on the right side was treated with ADP, the label should be corrected.
9. In Table S2-2, the labels "A1, A2..." may be instead of "R1, R2...".

Reviewer #2 (Remarks to the Author):

TREX (ATP-dependent Transcription-Export) complex is a regulator of mRNA export. One of the TREX complex components, UAP56 helicase, is a key factor in the assembly of the TREX complex. Upon ATP-binding to UAP56, TREX (ATP-TREX) recruit additional factors, thereby, facilitating mRNA export. In humans, a paralogue of UAP56, URH49, forms a distinct complex from the TREX complex, termed as AREX (Alternative-mRNA-export) complex. The composition of ATP-unbound AREX (apo-AREX) is different from that of TREX (apo-TREX). However, upon ATP-binding to URH49 of AREX, the AREX is remodeled and the component of ATP-bound AREX becomes homologous to those of ATP-bound TREX. UAP56 and URH49 selectively regulate the export of a specific subset of mRNAs, thus, the diversified mRNA export pathway by UAP56 and URH49 regulates the gene expression.

In the manuscript, Fujita et al. deal with the identification and characterization of the components of the apo-AREX complex and the mechanism of the different compositions between apo-TREX and apo-AREX by comparing the structures of UAP56 and URH49. The authors identified several apo-AREX components distinct from those of apo-TREX and showed that the newly identified apo-AREX components are indeed involved in the specific mRNA export. The author also showed that a single amino acid difference in the N-terminal domains of UAP56 and URH49 governs the composition of apo-TREX and the apo-AREX complexes. The author also showed that while the overall structures (in particular spatial arrangements of the N- and C- domains) of UAP56 and URH49 in their ATP-bound forms are homologous, the structures in their apo forms are different. These results imply that the specific single amino acid difference in their N-terminal domain of UAP56 and URH49 determines the export of a specific subset of mRNAs.

This study suggests that the formation of the two distinct mRNA export complexes depends on the distinct structural differences of homologous key helicases (UAP56 and URH49) in the complexes. This reviewer appreciates that the study presents mechanistic insights into the gene regulation by the distinct mRNA export complexes formation/composition (TREX and AREX). With that said, the rational explanation and discussion of the different structures of the apo UAP56 and URH49 are not provided. The structural differences between UAP56 and URH49 in their apo-forms are keys to the different mRNA exports. The structural comparison presented in the manuscript does not provide mechanistic insights into the distinct mRNA export complex formation (TREX or AREX).

The following points could be clarified before further consideration.

Major points

- 1) To enhance the accessibility of this study, the summary could be rewritten and reorganized. This reviewer thinks that the wording apo-complex in the abstract is confusing and that the potential readers would feel it difficult to follow the content because the description is too specific.
- 2) It is not clear why the spatial positions of the N- and C- domains of UAP56 and URH49 are different, even though the authors compare their structures. A single amino acid substitution in the N-terminal domains of UAP56 and URH49 can alter the composition of the TREX and AREX complex, and export a subset of mRNAs. This finding is one of the main ones in this manuscript. Thus, a rational explanation or pieces of evidence should be provided by the authors.
- 3) For the limited proteolysis of proteins (UAP56 and URH49) in the presence or absence of ADP. The authors assume that the ADP-bound and ATP-bound forms of UAP56 and URH49 are similar based on the data in Extended Data Fig.1 (page 9 lines 247-250). It seems that there are differences in composition for AREX-complex formation in the presence of +ADP and +AMP-PNP. Need to explain.
- 4) Related to 3). Did the authors do the limited proteolysis of proteins (UAP56 and URH49) in the presence of ATP? The same results could be obtained in the presence of ADP. (Is the remodeling ATP-hydrolysis dependent?)
- 5) Page 15: lines 414-416: "The structures of UAP56 and URH49 in apo-state showed the distinct configuration of the N- and C-domain due to different linker structures". Did the authors check the structure of UAP56 by themselves? The models in the public domain might not be well-refined, and

not well-modeled. Are the amino sequences of the linkers conserved? Please show the sequence alignments in the supplementary figures.

6) Page 15, lines 423-428. "The loop structure within the C-domain of ..., and prevent its own ATP binding" PAGE11: line 333. The data in Extended Data Fig8 G does not support the weaker ATP binding of URH49 than UAP56. The data presented show the difference is not significant. A quantitative kinetic analysis (ATP-hydrolysis) or affinity analysis would be required. The author could substitute amino acids in the loop and test the ATPase activity or ATP-binding.

Minor points

1) TEXT:

It is hard to follow the description. This reviewer suggests the authors rewrite and revise the text more clearly to enhance the accessibility to readers.

Page 2: lines 36-37, rephrase.

Page3: line69, rephrase

Page4: line 84 ~105. There are several sentences and wording and they are confusing. rewrite and rephrase.

Page6: line142, rephrase

Line158, rephrase

Page7: line188-189, rephrase

Line 2: line 203-210: Hard to follow. Rewrite.

Page8: line240, rephrase

Page9: line246-247, rephrase

Page 10: line295, rephrase.

PAGE11: line 333. The data in Extended Data Fig8 G does not support the weaker ATP binding of URH49 than UAP56. Is this significant? See major comments 6).

2) Figures

Extended Data Fig8G, H correct the labels of figures.

Reviewer #3 (Remarks to the Author):

In Fujita et al., the authors examine differences between UAP56 and URH49 and uncover new differential binding partners, differential functional regions, and different structural aspects. Although it is unclear how all of these differences add up to explain how these two proteins regulate the nuclear export of different mRNAs, the paper is very well executed and of general interest to the mRNA nuclear export community. I am generally in favour of publication provided that the authors address the following points:

1) For Figure 2D, some indication of the purity of the cytoplasmic fractionation should be provided (i.e. distribution of nuclear and cytoplasmic RNAs/proteins)?

2) When levels of a given mRNA decrease in the cytosol, does it go up in the nucleoplasm?

3) eCLIP for ILF3 exists (<https://www.nature.com/articles/s41586-020-2077-3>) and the author should determine whether ILF3 associates with mRNAs that are disproportionately exported by URH49.

4) An extended analysis of how distinct elements within URH49 and UAP56 are conserved (and how they differ between the two proteins) throughout vertebrates would be useful.

5) Some of the immunofluorescent images that are pseudocolored red are very hard to see. I would recommend that the authors show all single channel immunofluorescent images as grey-scale images to help the readers clearly interpret the data.

6) RUVBL1 and RUVBL2 are part of the R2TP complex which is thought to be involved in assembling multimeric complexes. This should be cited in the text. Are other components of the R2TP complex found in the URH49 precipitate?

Minor points:

- 1) UAP56 and URH49 are officially called DDX39B and DDX39A in humans – this should be stated at least once, just to help clarify confusions in the current literature.
- 2) Line 68: you may want to specify that “ZC11A” is an abbreviated form of “ZC3H11A”
- 3) Figure 3A some of the formatting of the numbers seems off (for example, the number “428” is split so that “42” and “8” are separated by a carriage return.)
- 4) Lines 217-218: “Thus, the mechanism of mRNA export appears to be different from that of circular RNA export.” Since the differential export of circular RNAs was not examined in this paper and there may be differences between cell lines, I would tone down this conclusion. Perhaps “how USP56 and URH49 differentially regulate the nuclear export of various circRNAs is not explained by differences in binding partners”.
- 5) Lines 225-226: “This result reflects that UAP56 and URH49 export distinct subsets of bulk mRNA substrates and do not the other.” This sentence is hard to parse and should be rewritten.
- 6) Line 387: “URH49 are required for..” should be “URH49 is required for..”
- 7) Line 404: a close bracket sign, “)”, is missing.

REPLY TO REVIEWER'S COMMENTS

Reviewer #1 (Remarks to the Author):

In this manuscript, using tandem-immunoprecipitation and mass spectrometry, Fujita *et al.* identified several novel key components together with URH49, a paralogue of helicase UAP56 in mammals, to form the apo-AREX complex. Although apo-AREX and apo-TREX complexes consist of distinct different components, they rebuilt into same ATP-complexes. The authors found that UAP56 and URH49 play key roles in dictating the formation of these two distinct apo-complexes. Surprisingly, only a single amino acid substitution in UAP56 or URH49 N-domain altered the formation and function of the apo-complexes. Further, using limited proteolysis assay, the authors claimed that UAP56 and URH49 exhibited different apo-structures, but with similar ADP binding conformation. They also solved the crystal structure of apo-URH49 Δ N41, which showed three different features compared to the apo-UAP56 Δ N42 structure. Based on these observations the authors proposed that UAP56 and URH49 form different complexes in the absence of ATP based on their diverse apo-conformation. This manuscript present mechanistic study that is important for understanding selective nuclear export of RNAs. The data are solid and the manuscript is overall clearly written. I have a couple of comments that the authors could consider to improve the study.

1. The authors performed immunoprecipitation with Flag-URH49 in 293 cells and detected many candidates for the apo-AREX components. Are the interactions of these new components with URH49 mediated by RNA? What is the function of the apo-AREX complex formed in the absence of ATP (or ADP). Considering that mRNA export requires ATP, it does not seem like it is important for this process. As this is the major finding of the work, the authors could discuss more, especially on the potential roles of this complex.

We appreciate your comment.

The RNase-treated nuclear extracts were used for immunoprecipitation. Thus, URH49 and the new AREX component can interact even in the absence of RNA. This point is clearly stated in the text (**Result line 110 and Discussion line 402**). We have described the potential role of the apo-TREX/AREX complex in mRNA processing, including splicing (**Discussion line 428**).

UAP56 is involved in splicing process by controlling spliceosome assembly through its ATPase and helicase activities, while whether URH49 is required for these activities remains unclear. Considering this, we propose the following model. UAP56 and URH49 function in the recognition and export of target mRNAs by forming their respective complexes.

Prior to splicing, UAP56 and URH49 are associated with their apo-complexes. The DEAD-box helicases to which UAP56 and URH49 belong generally bind to RNA independently of the RNA sequence. Chimeric mutants with swapped complex formation abilities exhibited a swap in target specificity between UAP56 and URH49, raising the possibility that their apo-complex components other than UAP56 and URH49 specify the selective regulation of RNA binding. Based on these results, it is likely that each apo-complex binds to specific RNA sequences or is guided by upstream processes (such as chromatin regulation), leading to subsequent interactions with the target mRNA by UAP56 and URH49. Subsequently, through the remodeling of each apo-complex into the ATP-TREX complex, the ATPase and helicase activities of UAP56 and URH49 are activated, leading to the splicing of the respective target pre-mRNA and, ultimately, mRNA export.

Indeed, we also observed that UAP56 and URH49 function in the splicing process alongside their respective apo-complexes.

We analyzed the eCLIP data for ILF3 and HNRNPM (ILF3: GEO: GSE91760, HNRNPM: GEO:

GSE91744) to determine whether ILF3 and HNRNPM associate with mRNAs that are disproportionately
exported by URH49. For each STAR-mapped bam dataset, binding reads per gene were counted using
Htseq count (doi:10.1093/bioinformatics/btu638) and normalized using Deseq2 (doi:10.18129/B9.bioc.D
ESeq2). Next, we analyzed whether there were differences in the binding levels of ILF3 and HNRNPM
between the target gene sets UAP56 and URH49. The target gene sets of UAP56 and URH49 were
defined based on the information (mRNAs with their cytoplasmic expression level specifically decreased
by 1.5 times or more by UAP56 and URH49 depletion) provided in the following paper
(doi:10.1091/mbc.E09-10-0913).

The analysis revealed no significant differences in the degree of binding between the UAP56-
or URH49-targets of ILF3 and HNRNPM (**For Reviewer's Figure (RFig).1, A-B**). To elucidate the RNA
binding sites of UAP56 and URH49, we performed Photo Activatable-Ribonucleoside-enhanced Cross
Linking and Immuno Precipitation (PAR-CLIP) of both helicases (unpublished data) and analyzed in the
same way. There was also no difference in the degree of UAP56 and URH49 binding between the two
groups (**RFig.1, C-D**).

We similarly analyzed the respective eCLIP data for ILF3 and HNRNPM and PAR-CLIP data for
UAP56 and URH49 for the newly defined target gene sets of UAP56 and URH49 using the following
protocol (The cytoplasmic RNA from UAP56 and URH49 knockdown cells was analyzed by RNAseq.
mRNAs with expression levels specifically decreased by two times or more by UAP56 and URH49
depletion were defined as new UAP56- or URH49-target mRNAs (unpublished data). The results
obtained from these analyses are consistent with those mentioned previously (**RFig.1, E-H**). These
results raise the possibility that UAP56 and URH49 (including the AREX complex) have the potential to
bind globally to their respective target mRNAs when evaluated across the entire transcriptome.

However, we observed that UAP56 and URH49 selectively bind to specific introns of their target
mRNAs and regulate the splicing of this site (**RFig.1, I**, GTPBP2 mRNA; UAP56 preferentially binds to
specific introns of this mRNA, as indicated by arrows, and UAP56 is required for splicing of this site.
C1ORF63 mRNA; URH49 preferentially binds to specific introns in the C1ORF63 mRNA and regulates
splicing at this site).

With the exception of certain introns, both UAP56 and URH49 commonly bind to many introns.
This suggests that UAP56 and URH49 bind to numerous mRNAs regardless of their respective targets.
When UAP56 or URH49 is depleted, the splicing of the respective target mRNAs is aberrant; therefore,
the target mRNAs are expected to remain in the nucleus.

The insights obtained from the eCLIP of ILF3 and HNRNPM are consistent with this model and
are also relevant to what we described as the potential role of the apo-TREX/AREX complex in mRNA
splicing (**Discussion line 435**). Because this aspect is beyond the scope of this study, it is pursued as
part of our future research endeavors.

2. FISH assay was performed to detected the nuclear poly(A) RNA accumulation in apo-AREX
components depleted U2OS cells. What is the reason that the authors selected the U2OS cells in the
FISH assay? They performed the immunoprecipitation in the 293 cells, dose the FISH assay can get
the same result as that in the U2OS cells?

We appreciate your comment.
There were two reasons for selecting U2OS cells for the RNA-FISH assay. First, compared to U2OS
cells, 293 cells require a larger quantity of siRNA and transfection reagents for siRNA-mediated
knockdown (please visit the information about the lipofectamine 2000 mediated siRNA transfection the
manufacture's web site below and in the site, you can easily see that 293 cells require more siRNA and
transfection than other cells in general. <https://www.thermofisher.com/jp/ja/home/life-science/cell->

culture/transfection/rnai-transfection/rnai-transfection-protocols.html). Second, in addition to higher
siRNA and transfection reagent requirements, the RNA-FISH assay is challenging in 293 cells
compared to U2OS cells due to the following reason. We investigated the transfection conditions in 293
cells and performed UAP56 knockdown. The RNA-FISH assay showed that, compared to the control
knockdown, poly(A)⁺ RNA (red signal) appeared to accumulate in the nuclei of UAP56 knockdown cells.
However, UAP56 knockdown cells shrank, making it difficult to distinguish between nuclear and
cytoplasmic poly(A)⁺ RNA (**RFig.2**). Therefore, we used U2OS cells to observe the observation of
poly(A)⁺ RNA localization.

3. The digested fragments of URH49ΔN41 and full length seems obtain different patterns, which
indicates the depleted N-terminal region may change the behaviour of URH49 against trypsin digestion.
The R1 and R4 bands are very weak, and the A1, A3 R3 include two bands, maybe the limited
proteolysis assay should be further optimized, such as incubating with less enzyme or shorter time.
Besides, the limited proteolysis assay is not the best way to detect the protein conformation in solution,
more evidence should be provided to confirm this observation.

We appreciate your comment.

We have realigned the positions of the marker molecular weights and revised the presentation of **Fig.**
**4A-D**. In addition, to objectively demonstrate the similarity between URH49 WT and URH49ΔN41 in terms
of the digested fragments (R2-like band and R2 band), we measured the mobility of the digested
fragments in **Fig. 4A-D** in comparison to the marker (**RFig.3, RFig.3.xlsx**). For comparison, we also
measured the similarity between A1-like band and A1 band of UAP56 WT and UAP56ΔN42, respectively.
The results indicate that the positions of the bands (R2-like band and R2 band) derived from URH49 WT
and URH49ΔN41 are similar. We appreciate this comment, which has helped us rectify any potentially
misleading expressions.

Additionally, we conducted experiments involving varying enzyme concentrations and shorter
incubation times but were unable to observe limited digestion products other than A1-A3 and R1-R4 from
UAP56ΔN42 and URH49ΔN41 (**RFig.4**). Although a limited proteolysis assay may not be the best method
to detect protein conformation in solution, in this study, we believe it is sufficient to demonstrate the
following three points: 1) Under the conditions with the absence of ATP, UAP56ΔN42 and URH49ΔN41
indeed exhibit structural differences in solution. 2) URH49 C223VΔN41 shows UAP56ΔN42-like structural
features indicating the significance of the differences between UAP56 224V and URH49 223C in the
structural variances observed in the solution under ATP deficient conditions. 3) Under ATP loaded
conditions, UAP56 and URH49 adopted a UAP56-like structure in solution. In the main text, we have also
mentioned the need for future analyses of the protein conformations of UAP56 and URH49 in solution
(**Discussion line 494**). In addition to the above analysis, we also performed a Small Angle X-ray
Scattering analysis (SEC-SAXS), to detect the protein conformation of UAP56 and URH49 in solution.
However, obtaining sufficiently precise data is challenging. This is because this analysis requires highly
purified UAP56ΔN42 and URH49ΔN41 in large quantities, along with extensive optimization of conditions
for the analysis.

4. In Extended Data Fig.7A, the authors showed the URH49ΔN41(-ADP) was digested into two
separate fragments R1 and R2, which was different from the other three samples. There seems no
solid evidence to support this model.

We appreciate your comment.

As suggested, we eliminated the model (**Extended Data Fig. 7A** in original paper) because there was
no experimental evidence for the order in which the degradation products occurred, as in this model.

5. The two RecA domains within UAP56 do not adopt fixed relative orientation in solution, and the
crystal structure reported here reflects just one state of URH49 in solution. So, more evidences are
required to support the conclusion that the apo-structures of UAP56 and URH49 make them to
integrate into separate complexes.

We appreciate your comment.

As pointed out, the crystal structure generally reflects one of the possible structures in solution. We found
that the RecA1 and RecA2 domains of UAP56 and URH49 differ by 32 °in their crystal structures (**Fig.**
**5A**) but have not yet verified the extent to which the two domains are different in solution. Further analysis
is required to verify this issue, which has been added to the revised text (**Discussion line 494**). We have
also added that unidentified factors other than the observed differences in the crystal structures of UAP56
and URH49 may affect the difference in complex formation between UAP56 and URH49 (**Discussion**
**line 497**).

To show that UAP56 and URH49 predominantly adopt different conformations in solution, we
performed limited digestion of purified UAP56 and URH49 in solution and found that the products of their
limited digestion differed under apo-conditions in which no ADP was added (**Fig. 4C**). This suggests that
a significant portion of the apo-structures of UAP56 and URH49 may differ in solution. In addition, the
limited digestion products of the URH49 C223 mutant, which exhibited a switch in complex formation
from the apo-AREX to the apo-TREX complex, showed highly similar patterns to those of the limited
digestion products of UAP56 under apo-conditions (**Fig. 4C**). These observations suggest that the
difference in the apo-structures of UAP56 and URH49 in solution is a major determinant of complex
formation.

To verify whether the apo-structures of UAP56 and URH49 are crucial for complex formation, it
is necessary to examine the structures of UAP56 and URH49 in solution within the complex (e.g., cryo-
EM analysis of the complex) and to observe how complex formation occurs. These analyses are
important for understanding the mechanism of complex formation between UAP56 and URH49. This
has been clearly stated in the text (**Discussion line 499**). Based on this, since the current study
focused on the identification and functional analysis of novel components of the AREX complex, in
addition to the possible major factors of complex differentiation, we would like to proceed with that as a
separate study.

6. It is intriguing that no structural difference was observed with the key amino acid (URH49 C223 and
UAP56 V224) that determines the different apo-complex of UAP56 and URH49. The author proposed
that the difference in the loop structure might be caused by this amino acid. It would be great if they
could use molecular modelling to examine this possibility.

We appreciate your comment.

As you mentioned, we created a model based on homology modeling using Modeller
(<https://salilab.org/modeller/>) (**RFig.5**). To estimate the plausibility of the structural model obtained by
the analysis, a structural model of URH49 was created based on the UAP56 crystal structure (1XTI)
(upper panel, URH49Δ41 model). The resulting URH49 structure model is almost identical to the
UAP56 crystal structure (1XTI) (RMSD = 0.22 Å), unlike the structure model (Fr48) derived from the
actual URH49 crystal structure. This result suggests that in UAP56 and URH49, where the total amino

acid homology is high, is unable to build a structural model from the template by molecular modeling;
even if the URH49 Δ 41 C223V model is created from the URH49 structural model (Fr48), the structure
of the URH49 Δ 41 C223V model was similar to that of the URH49 structural model (Fr48) (lower panel).
Therefore, molecular modeling could not validate whether UAP56 V224 and URH49 C223 were key to
the structural differences between the two.

Therefore, to identify the residues that could potentially cause global structural differences
between UAP56 and URH49, we exhaustively calculated the alteration of all backbone dihedral angles
between the two molecules and detected the critical residues whose local conformations differed
significantly (dihedral angles greater than 120°, as shown in magenta in **Fig. 5C**, see also **Table.S3**).

Some residues were located in the linker region of UAP56 and URH49 (in the linker region:
UAP56-254E, -257 L, URH49-253E, and -256 L). This suggests that the differences in the arrangement
of these residues may be the cause of the structural differences between UAP56 and URH49 in linker
region. Among these amino acid residues, the amino acids closest to UAP56 V224 and URH49 C223
were UAP56 D245 and URH49 D244, located at the UAP56 243-245 linker and URH49 242-244 linker,
respectively (**Fig. 5C**). In addition, linker sites are generally susceptible to structural changes. We
examined the amino acid residues that interact with UAP56 V224 and URH49 C223 on each linker and
found UAP56 M243 and URH49 M242 residues (**Fig. 5D**). We observed that UAP56 M243 and URH49
M242 directly interacted with UAP56 V224 and URH49 C223, respectively, altering the spatial
arrangement of UAP56 M243 and URH49 M242, which in turn altered the arrangement of UAP56 D245
and URH49 D244 (**Fig. 5D**). These spatial arrangements influence subsequent linkers (UAP56 243-245
linker and URH49 242-244 linker) and interdomain linkers and are expected to eventually contribute to
the C-domain arrangement. Thus, URH49 C223 and UAP56 V224 may be the key amino acids
responsible for their structural differences. This point was stated in the text (**Result line 344**).

7. It would be important for the authors to at least discuss how the difference in their apo-complexes
determines the substrate specificity of UAP56 and URH49.

We appreciate your comment.

As answered to comment 1, we added to the potential role of the apo-AREX complex in the discussion
section (**Discussion line 428**).

8. In Extended Data Fig.7C, I guess the sample on the right side was treated with ADP, the label should
be corrected.

We appreciate your comment.

The notation in this part was incorrect and has been corrected (**Extended Data Fig.9B**).

9. In Table S2-2, the labels “A1, A2...” may be instead of “R1, R2...”.

We appreciate your comment.

The notation in this part was incorrect and has been corrected (**Table S2-2**).

**Reviewer #2 (Remarks to the Author):**

TREX (ATP-dependent Transcription-Export) complex is a regulator of mRNA export. One of the
TREX complex components, UAP56 helicase, is a key factor in the assembly of the TREX complex.
Upon ATP-binding to UAP56, TREX (ATP-TREX) recruit additional factors, thereby, facilitating mRNA
export. In humans, a paralogue of UAP56, URH49, forms a distinct complex from the TREX complex,
termed as AREX (Alternative-mRNA-export) complex. The composition of ATP-unbound AREX (apo-
AREX) is different from that of TREX (apo-TREX). However, upon ATP-binding to URH49 of AREX,
the AREX is remodeled and the component of ATP-bound AREX becomes homologous to those of
ATP-bound TREX. UAP56 and URH49 selectively regulate the export of a specific subset of mRNAs,
thus, the diversified mRNA export pathway by UAP56 and URH49 regulates the gene expression.

In the manuscript, Fujita *et al.* deal with the identification and characterization of the components of the
apo-AREX complex and the mechanism of the different compositions between apo-TREX and apo-
AREX by comparing the structures of UAP56 and URH49. The authors identified several apo-AREX
components distinct from those of apo-TREX and showed that the newly identified apo-AREX
components are indeed involved in the specific mRNA export. The author also showed that a single
amino acid difference in the N-terminal domains of UAP56 and URH49 governs the composition of apo-
TREX and the apo-AREX complexes. The author also showed that while the overall structures (in
particular spatial arrangements of the N- and C-domains) of UAP56 and URH49 in their ATP-bound
forms are homologous, the structures in their apo forms are different. These results imply that the
specific single amino acid difference in their N-terminal domain of UAP56 and URH49 determines the
export of a specific subset of mRNAs.

This study suggests that the formation of the two distinct mRNA export complexes depends on the
distinct structural differences of homologous key helicases (UAP56 and URH49) in the complexes.
This reviewer appreciates that the study presents mechanistic insights into the gene regulation by the
distinct mRNA export complexes formation/composition (TREX and AREX). With that said, the
rational explanation and discussion of the different structures of the apo UAP56 and URH49 are not
provided. The structural differences between UAP56 and URH49 in their apo-forms are keys to the
different mRNA exports. The structural comparison presented in the manuscript does not provide
mechanistic insights into the distinct mRNA export complex formation (TREX or AREX).

The following points could be clarified before further consideration.

Major points

1) To enhance the accessibility of this study, the summary could be rewritten and reorganized. This
reviewer thinks that the wording apo-complex in the abstract is confusing and that the potential readers
would feel it difficult to follow the content because the description is too specific.

We appreciate your comment.

We rewrote the entire Summary section.

2) It is not clear why the spatial positions of the N- and C-domains of UAP56 and URH49 are different,
even though the authors compare their structures. A single amino acid substitution in the N-terminal
domains of UAP56 and URH49 can alter the composition of the TREX and AREX complex, and export

a subset of mRNAs. This finding is one of the main ones in this manuscript. Thus, a rational
explanation or pieces of evidence should be provided by the authors.

We are grateful for your comment.

As mentioned, we created a model based on homology modeling using Modeller
(<https://salilab.org/modeller/>) (**For Reviewer's Figure (RFig).5**). To estimate the plausibility of the
structural model obtained by the analysis, a structural model of URH49 was created based on the
UAP56 crystal structure (1XTI) (upper panel, URH49 Δ 41 model). The resulting URH49 structure model
is almost identical to the UAP56 crystal structure (1XTI) (RMSD = 0.22 Å), unlike the structure model
(Fr48) derived from the actual URH49 crystal structure. This result suggests that in UAP56 and URH49,
where the total amino acid homology is high, is unable to build a structural model from the template by
molecular modeling; even if the URH49 Δ 41 C223V model is created from the URH49 structural model
(Fr48), the structure of the URH49 Δ 41 C223V model was similar to that of the URH49 structural model
(Fr48) (lower panel). Therefore, molecular modeling could not validate whether UAP56 V224 and
URH49 C223 were key to the structural differences between the two.

Therefore, to identify the residues that could potentially cause global structural differences
between UAP56 and URH49, we exhaustively calculated the alteration of all backbone dihedral angles
between the two molecules and detected the critical residues whose local conformations differed
significantly (dihedral angles greater than 120°, as shown in magenta in **Fig. 5C**, see also **Table.S3**).

Some residues were located in the linker region of UAP56 and URH49 (in the linker region:
UAP56-254E, -257 L, URH49-253E, and -256 L). This suggests that the differences in the arrangement
of these residues may be the cause of the structural differences between UAP56 and URH49 in linker
region. Among these amino acid residues, the amino acids closest to UAP56 V224 and URH49 C223
were UAP56 D245 and URH49 D244, located at the UAP56 243-245 linker and URH49 242-244 linker,
respectively (**Fig. 5C**). In addition, linker sites are generally susceptible to structural changes. We
examined the amino acid residues that interact with UAP56 V224 and URH49 C223 on each linker and
found UAP56 M243 and URH49 M242 residues (**Fig. 5D**). We observed that UAP56 M243 and URH49
M242 directly interacted with UAP56 V224 and URH49 C223, respectively, altering the spatial
arrangement of UAP56 M243 and URH49 M242, which in turn altered the arrangement of UAP56 D245
and URH49 D244 (**Fig. 5D**). These spatial arrangements influence subsequent linkers (UAP56 243-245
linker and URH49 242-244 linker) and interdomain linkers and are expected to eventually contribute to
the C-domain arrangement. Thus, URH49 C223 and UAP56 V224 may be the key amino acids
responsible for their structural differences. This point was stated in the text (**Result line 344**).

3) For the limited proteolysis of proteins (UAP56 and URH49) in the presence or absence of ADP, the
authors assume that the ADP-bound and ATP-bound forms of UAP56 and URH49 are similar based on
the data in Extended Data Fig.1 (page 9 lines 247-250). It seems that there are differences in
composition for AREX-complex formation in the presence of +ADP and +AMP-PNP. Need to explain.

We appreciate your comment.

UAP56 binds AMP-PNP with affinities at least 10 times lower than that of ATP and binds ADP with
similar affinities as ATP (doi:10.1128/MCB.01341-07). Therefore, in the presence of AMP-PNP, it is
possible that a part of apo-URH49 molecules do not bind to AMP-PNP. Consequently, we hypothesized
that with the addition of AMP-PNP, both the ATP-TREX (URH49) complex and partially the apo-AREX
complex may coexist in the immunoprecipitates of URH49, making it appear different from the ATP-
bound form of URH49. To prove this experimentally, we performed immunoprecipitation of URH49 in
the presence of excess AMP-PNP. Under conditions of excess AMP-PNP, immunoprecipitation of

URH49 revealed coprecipitates closely resembling those of ATP-bound URH49 (see **Extended Data**
**Fig.1B**). This point has been stated in the text (**Result line 121**).

4) Related to 3). Did the authors do the limited proteolysis of proteins (UAP56 and URH49) in the
presence of ATP? The same results could be obtained in the presence of ADP. (Is the remodeling
ATP-hydrolysis dependent?)

We appreciate your comment.

We have added additional information regarding the limited proteolysis assay in the presence of ATP.
The limited proteolysis of the proteins (UAP56 and URH49) in the presence of ATP closely resembled
those obtained when ADP was added (**Extended Data Fig. 9A**). This suggests that ATP-dependent
structural changes in URH49 are likely to be ATP hydrolysis-independent. These findings are consistent
with the notion that the complex remodeling of URH49 is induced by the addition of ADP (**Extended**
**Data Fig. 1A**). This information has been added to the revised text (**line 280**).

5) Page 15: lines 414-416: “The structures of UAP56 and URH49 in apo-state showed the distinct
configuration of the N- and C-domain due to different linker structures”. Did the authors check the
structure of UAP56 by themselves? The models in the public domain might not be well-refined, and
not well-modeled. Are the amino sequences of the linkers conserved? Please show the sequence
alignments in the supplementary figures.

We appreciate your comment.

We did not crystallize UAP56, but analyzed the crystal structure of UAP56 (1XTI) reported in the PDB
database. The validity values of this crystal structure model are as follows and are considered
sufficiently certain. Resolution: 1.95 Å, R-Value Free:0.257, R-Value Work:0.218 (1XTI was obtained
from the PDB database, <https://www.rcsb.org/structure/1xti>). The amino acid sequence of the linker
region between the N and C domains is conserved in UAP56 and URH49. In the original submission,
we could not effectively highlight the linker region. Therefore, we have made a change to address this
issue.

The crystal structure generally reflects one of the possible structures in solution. We found that
the RecA1 and RecA2 domains of UAP56 and URH49 differ by 34.82 °in their crystal structures (**Fig.**
**5A**) but have not yet verified the extent to which the two domains are different in solution. Further
analysis is required to verify this issue, which has been added to the revised text (**Discussion line**
**494**). We have also added that unidentified factors other than the observed differences in the crystal
structures of UAP56 and URH49 may have affected the differences in complex formation between
UAP56 and URH49.

The amino acid sequence of the linker region was conserved, but was not effectively
highlighted in the previous alignment. We corrected this in the revised manuscript (**Extended Data Fig.**
**4**). This point was stated in the text (**Result line 371**).

6) Page 15, lines 423-428. “The loop structure within the C-domain of ..., and prevent its own ATP
binding”. Page 11: line 333. The data in Extended Data Fig. 8G does not support the weaker ATP
binding of URH49 than UAP56. The data presented show the difference is not significant. A
quantitative kinetic analysis (ATP-hydrolysis) or affinity analysis would be required. The author could
substitute amino acids in the loop and test the ATPase activity or ATP-binding.

We appreciate your comment.
It is important to note that quantitative analysis is required to confirm these observations. In addition,
based on the experimental results described below, we believe that ongoing research will be necessary
to confirm the function of URH49 C-loop in the ATP-binding of URH49. Therefore, the results of the
verification of ATP-binding and helicase activities of UAP56 and URH49 (**Extended Data Fig. 8G, H** in
the original paper) were removed from the revised manuscript, and the potential role of URH49 C-loop
was described in Discussion (**line 502**). The removal of these data has no impact on the essence of this
manuscript. Next, we describe the results of the experiments conducted to investigate the function of
the URH49 C-loop.

We generated two mutants of URH49, URH49 Δ C-loop mutant with a deletion of the C-loop
and URH49 C-loop mutant with eleven amino acid substitutions in the C-loop, replacing them with
adenine, to compare their ATP binding with the wild-type URH49. The obtained results did not meet our
expectations probably because the C-loop region exists as a conserved motif (motif V in **Extended**
**Data Fig.4**) which is essential for ATP binding and ATPase activity in DEAD-box helicases to which
UAP56 and URH49 belong (doi:10.1111/febs.13198). Therefore, altering the C-loop region by the
deletion or mutation could lead to difficulties in proper folding of the recombinant protein produced in *E.*
*coli* or a decrease in the stability of the folded protein, making it challenging to accurately evaluate ATP
binding under experimental conditions. For these reasons, further investigation is required to determine
whether the C-loop region of URH49 directly inhibits ATP binding. We believe that ongoing research will
be necessary to confirm this.

Minor points

1) TEXT:

It is hard to follow the description. This reviewer suggests the authors rewrite and revise the text more
clearly to enhance the accessibility to readers.

Page 2: lines 36-37, rephrase.

We rewrote the entire Summary section.

Page 3: line69, rephrase

We rephrased the following (**line 67**).

Thus, the TREX complex is remodeled from the ATP-unbound TREX complex to the ATP-bound TREX
complex via ATP binding to UAP56.

Page 4: line 84 ~105. There are several sentences and wording and they are confusing. rewrite and
rephrase.

We rephrased two paragraphs so that these were easier for the reader to read. These paragraphs were
in **lines 83 ~102** of the revised manuscript.

Page 6: line142, rephrase. Line158, rephrase

We rephrased the followings.

Line 148, To confirm this, we generated cell lines stably expressing FLAG-RUVBL1, RUVBL2, ILF2, ILF3, and HNRNPM and investigated their interactions with URH49.

Line 163, NF90, a truncated isoform of ILF3, also interacts with ILF2.

Page 7: line188-189, rephrase. Line 2: line 203-210: Hard to follow. Rewrite.

We rephrased the followings.

Line 196, To identify the region(s) within UAP56 and URH49 responsible for forming different apo-complexes, we constructed plasmids expressing various mutants of UAP56 and URH49, in which different regions were swapped. We examined the formation of apo-complexes.

Line 212, To further investigate the potential contribution of amino acid differences other than UAP56-V224 and URH49-C223 to their distinct complex formation, we generated a mutant termed "UAP56 N-core C224V." In this mutant, the N-domain of UAP56, excluding V224, was replaced with the N-domain of URH49. This mutant lost the ability to form the apo-AREX complex but retained the ability to form the apo-TREX complex.

Page 8: line240, rephrase

We rephrased the followings.

Line 255, However, upon ATP binding, the N- and C-domains undergo rearrangement into similar closed structures driven by interactions with ATP.

Page 9: line246-247, rephrase

We rephrased the followings.

Line 260, These results led us to hypothesize that the structures of UAP56 and URH49 in their apo- and ATP-bound states determine the formation of their apo- and ATP complexes.

Page 10: line 295, rephrase.

We rephrased the followings.

Line 312, Structural differences between the apo-UAP56 and URH49.

Page 11: line 333. The data in Extended Data Fig. 8G does not support the weaker ATP binding of URH49 than UAP56. Is this significant? See major comments 6).

We appreciate your comment.

We are grateful for addressing the labeling errors in the previous data. The data in Extended Data Fig.
8G in the original paper were removed. The removal of this data has no impact on the essence of this
manuscript.

2) Figures

Extended Data Fig. 8G, H correct the labels of figures.

We appreciate your comment.

The data in Extended Data Fig. 8G, H in the original paper were removed. The reason why these data
were removed is described above.

**Reviewer #3 (Remarks to the Author):**

In Fujita *et al.*, the authors examine differences between UAP56 and URH49 and uncover new
differential binding partners, differential functional regions, and different structural aspects. Although it
is unclear how all of these differences add up to explain how these two proteins regulate the nuclear
export of different mRNAs, the paper is very well executed and of general interest to the mRNA nuclear
export community. I am generally in favour of publication provided that the authors address the
following points:

1) For Figure 2D, some indication of the purity of the cytoplasmic fractionation should be provided (i.e.
distribution of nuclear and cytoplasmic RNAs/proteins)?

We appreciate your comment.

We conducted RT-PCR on substrates in the nucleus and the cytoplasm to confirm the cellular
fractionation of the nucleus and the cytoplasm. We have presented the data in **Extended Data Fig. 3E**,
which has been added to the revised text (**line 178**).

2) When levels of a given mRNA decrease in the cytosol, does it go up in the nucleoplasm?

We appreciate your comment.

We extracted nuclear RNA from cells in which UAP56 or URH49 was knocked down and examined the
expression levels of their respective target mRNAs. The target mRNAs of UAP56 and URH49, which
showed decreased expression in the cytoplasm, also showed decreased expression in the nucleus (**For
Reviewer's Figure (RFig).6**).

In nuclear RNA metabolism, factors involved in mRNA export (such as ALYREF) and factors
involved in degradation (such as MTR4) competitively bind to the mRNA, thereby determining whether
the target mRNA is exported or degraded (DOI:10.15252/embj.201696139). Based on this, we
hypothesized that mRNA undergoing export blockage may be degraded by nuclear exosomes. To test
this possibility, we performed double knockdown of either UAP56 or URH49 and the nuclear exosome
component RRP45. We observed an increase in the nuclear levels of UAP56's target mRNAs MT2A and
HN1. However, for other mRNAs, there was no increase in the nuclear levels, even with double
knockdown.

As a result, it has been suggested that at least some of the target mRNAs of UAP56 (e.g., MT2A
and HN1) are subject to degradation by nuclear exosomes due to export inhibition. Several possibilities
exist for other target mRNAs; however, in our current investigation, we were unable to reach definitive
conclusions for the following reasons: **1.** Some mRNAs may not accumulate in the nucleus because they
undergo RNA degradation independent of RRP45. **2.** Some mRNAs may be subject to degradation by
RRP45; however, the knockdown of RRP45 in this study may not have been sufficient to observe a
significant impact. **3.** The knockdown of UAP56 or URH49 may lead to suppression at earlier stages,
such as during transcription, which could affect the fate of other mRNAs.

Further research is needed to explore these possibilities and gain a deeper understanding of
the fate of different target mRNAs in the context of UAP56 and URH49 function. The mechanism of fate
determination of each target mRNA is also of interest but is beyond the scope of this study and has not
been analyzed further.

3) eCLIP for ILF3 exists (<https://www.nature.com/articles/s41586-020-2077-3>) and the author should
determine whether ILF3 associates with mRNAs that are disproportionately exported by URH49.

We appreciate your comment.

We analyzed the eCLIP data for ILF3 and HNRNPM (ILF3: GEO: GSE91760; HNRNPM: GEO:
GSE91744). For each STAR-mapped bam dataset, binding reads per gene were counted using Htseq
count (doi:10.1093/bioinformatics/btu638) and normalized using Deseq2 (doi:10.18129/B9.bioc.DESeq2).
Next, we analyzed whether there were differences in the binding levels of ILF3 and HNRNPM between
the target gene sets UAP56 and URH49. The target gene sets of UAP56 and URH49 were defined based
on the information (mRNAs with their cytoplasmic expression level specifically decreased by 1.5 times or
more by UAP56 and URH49 depletion) provided in the following paper (doi:10.1091/mbc.E09-10-0913).

The analysis revealed no significant differences in the degree of binding between the UAP56-
or URH49-targets of ILF3 and HNRNPM (**RFig.1, A-B**). To elucidate the RNA binding sites of UAP56 and
URH49, we performed Photo Activatable-Ribonucleoside-enhanced Cross Linking and Immuno
Precipitation (PAR-CLIP) of both helicases (unpublished data) and analyzed in the same way. There was
also no difference in the degree of UAP56 and URH49 binding between the two groups (**RFig.1, C-D**).

We similarly analyzed the respective eCLIP data for ILF3 and HNRNPM and PAR-CLIP data for
UAP56 and URH49 for the newly defined target gene sets of UAP56 and URH49 using the following
protocol (The cytoplasmic RNA from UAP56 and URH49 knockdown cells was analyzed by RNAseq.
mRNAs with expression levels specifically decreased by two times or more by UAP56 and URH49
depletion were defined as new UAP56- or URH49-target mRNAs (unpublished data). The results
obtained from these analyses are consistent with those mentioned previously (**RFig.1, E-H**). These
results raise the possibility that UAP56 and URH49 (including the AREX complex) have the potential to
bind globally to their respective target mRNAs when evaluated across the entire transcriptome.

However, we observed that UAP56 and URH49 selectively bind to specific introns of their target
mRNAs and regulate the splicing of this site (**RFig.1, I**, GTPBP2 mRNA; UAP56 preferentially binds to
specific introns of this mRNA, as indicated by arrows, and UAP56 is required for splicing of this site.
C1ORF63 mRNA; URH49 preferentially binds to specific introns in the C1ORF63 mRNA and regulates
splicing at this site).

With the exception of certain introns, both UAP56 and URH49 commonly bind to many introns.
This suggests that UAP56 and URH49 bind to numerous mRNAs regardless of their respective targets.
When UAP56 or URH49 is depleted, the splicing of the respective target mRNAs is aberrant; therefore,
the target mRNAs are expected to remain in the nucleus.

The insights obtained from the eCLIP of ILF3 and HNRNPM are consistent with this model and
are also relevant to what we described as the potential role of the apo-TREX/AREX complex in mRNA
splicing (**Discussion line 435**). However, we require a more detailed analysis to define this finding,
which is different from the focus of the current study, and consider reporting it in a subsequent paper.

4) An extended analysis of how distinct elements within URH49 and UAP56 are conserved (and how
they differ between the two proteins) throughout vertebrates would be useful.

We appreciate your comment.

As you have pointed out, we selected 10 vertebrate species and aligned the amino acid sequences of
their respective UAP56 and URH49 homologs (**Extended Data Fig.6, 7**). UAP56-V224 and URH49-
C223 were conserved across these organisms. This finding suggests a potential implication of UAP56-
V224 and URH49-C223 in the evolutionary functional divergence of UAP56 and URH49. We have
added this information to the main text (**line 218**).

5) Some of the immunofluorescent images that are pseudo-colored red are very hard to see. I would recommend that the authors show all single channel immunofluorescent images as grey-scale images to help the readers clearly interpret the data.

We appreciate your comment.

The incorrect notation has been corrected (**Fig.2,3, Extended Data Fig.3.**).

6) RUVBL1 and RUVBL2 are part of the R2TP complex which is thought to be involved in assembling multimeric complexes. This should be cited in the text. Are other components of the R2TP complex found in the URH49 precipitate?

We appreciate your comment.

In light of the references ([doi:10.1016/j.bbamcr.2011.08.016](https://doi.org/10.1016/j.bbamcr.2011.08.016)) confirming that RUVBL1 and RUVBL2 are part of the R2TP complex, we have included this information in the revised manuscript (**Discussion, line 414**).

Furthermore, our investigation of whether the R2TP complex ([GO:0097255](https://www.ebi.ac.uk/QuickGO/GTerm?id=GO:0097255), <https://www.ebi.ac.uk/QuickGO/GTerm?id=GO:0097255>), the Ino80 complex ([GO:0031011](https://www.ebi.ac.uk/QuickGO/GTerm?id=GO:0031011), <https://www.ebi.ac.uk/QuickGO/GTerm?id=GO:0031011>) and the prefoldin-like complex ([GO:1990062](https://www.ebi.ac.uk/QuickGO/GTerm?id=GO:1990062), <https://www.ebi.ac.uk/QuickGO/GTerm?id=GO:1990062>), are present in the URH49 precipitate, using both references and the DAVID database, failed to find any. The absence of these factors, except for RUVBL1 and RUVBL2, suggests that RUVBL1 and RUVBL2 may form a separate complex with URH49, independent of the mentioned complexes. We have added this point to the Discussion (**Discussion, line 419**).

Minor points:

1) UAP56 and URH49 are officially called DDX39B and DDX39A in humans – this should be stated at least once, just to help clarify confusions in the current literature.

We appreciate your comment.

The notation in this part was incorrect and has been corrected (**line 59, 74**).

2) Line 68: you may want to specify that “ZC11A” is an abbreviated form of “ZC3H11A”

We appreciate your comment.

The notation in this part was incorrect and has been corrected (**line 66**).

The abbreviation of “ZC11A” has been corrected to “ZC3H11A”.

3) Figure 3A some of the formatting of the numbers seems off (for example, the number “428” is split so that “42” and “8” are separated by a carriage return.)

We appreciate your comment.

The notation in this part was incorrect and has been corrected (**Fig.3A**).

4) Lines 217-218: “Thus, the mechanism of mRNA export appears to be different from that of circular RNA export.” Since the differential export of circular RNAs was not examined in this paper and there may be differences between cell lines, I would tone down this conclusion. Perhaps “how USP56 and URH49 differentially regulate the nuclear export of various circRNAs is not explained by differences in binding partners”.

We appreciate your comment.

We agree with you. Our arguments were a little too strong. Thus, we rephrased the followings (**line 230**).

How UAP56 and URH49 differentially regulate the export of various circRNAs is unlikely to be explained by differences in binding partners involved in mRNA export.

5) Lines 225-226: “This result reflects that UAP56 and URH49 export distinct subsets of bulk mRNA substrates and do not the other.” This sentence is hard to parse and should be rewritten.

We appreciate your comment.

We rephrased the followings (**line 239**).

These results reflect that UAP56 and URH49 are involved in the export of distinct subsets of mRNA substrates, and there are likely non-redundant mRNA substrates that are specific to each of them.

6) Line 387: “URH49 are required for...” should be “URH49 is required for...”

We appreciate your comment.

The notation in this part was incorrect and has been corrected (**line 455**).

7) Line 404: a close bracket sign, “)”, is missing.

We appreciate your comment.

The notation in this part was incorrect and has been corrected (**line 471**).

REVIEWERS' COMMENTS

Reviewer #1 (Remarks to the Author):

The authors have responded to my comments in a satisfied way, and I do not have further concerns. I think now the manuscript should be accepted for publication.

Reviewer #2 (Remarks to the Author):

The authors have addressed most of the concerns raised by this reviewer. The reviewer appreciates the authors' efforts to explain the different spatial positioning of the N- and C-domains between UAP56 and URH49 in the revised manuscript (Page 12, line 344 to Page 13, line 379). However, much of the description is speculative, and this reviewer believes that the hypothesis may be challenging for other researchers to accept. The concern remains, and the question persists regarding how the structural differences between UAP56 and URH49 influence the formation of different apo complexes (TREX and AREX).

The reviewer highly appreciates the identification and functional analysis of the novel components of the AREX complex and the subsequent characterization of UAP56 and URH49 as possible major factors in the formation of distinct complexes for TREX and AREX. The authors have clearly demonstrated that a single amino acid difference between UAP56 and URH49 (UAP56-V224 and URH49-C223) indeed impacts the formation of distinct apo-complexes and regulates the export of specific groups of mRNAs. The authors have also shown that the structures probed by trypsin-limited digestion strongly suggest different spatial positioning of the N- and C-domains between UAP56 and URH49.

To improve this manuscript and enhance its accessibility to readers, this reviewer suggests that the description on pages 12-13 should be shortened and simply mention that the clarification of the mechanism awaits further structural and biochemical studies and/or move this information to the supplementary materials.

Minor Issue:

Figure 5C: Change "linker" to "loop"  D245 (in the 243-245 loop) or D244 (in the 242-244 loop). This description is confusing. Furthermore, the description on Page 12-13 should be corrected accordingly.

Figure 5D: The characters showing distances in the panels are bold. Please correct them.

Reviewer #3 (Remarks to the Author):

The authors have addressed all my concerns.

REPLY TO REVIEWER'S COMMENTS

Reviewer #1 (Remarks to the Author):

The authors have responded to my comments in a satisfied way, and I do not have further concerns. I think now the manuscript should be accepted for publication.

We appreciate your comment.

Reviewer #2 (Remarks to the Author):

The authors have addressed most of the concerns raised by this reviewer. The reviewer appreciates the authors' efforts to explain the different spatial positioning of the N- and C-domains between UAP56 and URH49 in the revised manuscript (Page 12, line 344 to Page 13, line 379). However, much of the description is speculative, and this reviewer believes that the hypothesis may be challenging for other researchers to accept. The concern remains, and the question persists regarding how the structural differences between UAP56 and URH49 influence the formation of different apo complexes (TRES and AREX).

The reviewer highly appreciates the identification and functional analysis of the novel components of the AREX complex and the subsequent characterization of UAP56 and URH49 as possible major factors in the formation of distinct complexes for TRES and AREX. The authors have clearly demonstrated that a single amino acid difference between UAP56 and URH49 (UAP56-V224 and URH49-C223) indeed impacts the formation of distinct apo-complexes and regulates the export of specific groups of mRNAs. The authors have also shown that the structures probed by trypsin-limited digestion strongly suggest different spatial positioning of the N- and C-domains between UAP56 and URH49.

To improve this manuscript and enhance its accessibility to readers, this reviewer suggests that the description on pages 12-13 should be shortened and simply mention that the clarification of the mechanism awaits further structural and biochemical studies and/or move this information to the supplementary materials.

We appreciate your comment.

As mentioned in your comments, we shortened pages 12-13 and mentioned that the clarification of the mechanism awaits further structural and biochemical studies in the final remarks.

Minor Issue:

Figure 5C: Change "linker" to "loop"  D245 (in the 243-245 loop) or D244 (in the 242-244 loop). This description is confusing. Furthermore, the description on Page 12-13 should be corrected accordingly.

We appreciate your comment.

We have edited Figure 5C and the main text according to your suggestion.

Figure 5D: The characters showing distances in the panels are bold. Please correct them.

We appreciate your comment.

We have edited Figure 5D and the main text according to your suggestion.

**Reviewer #3 (Remarks to the Author):**
The authors have addressed all my concerns.
We appreciate your comment.